# Airborne coherent wind lidar measurements of the momentum flux profile from orographically induced gravity waves

Benjamin Witschas[1], Sonja Gisinger[1], Stephan Rahm[1], Andreas Dörnbrack[1], David C. Fritts[2], and Markus Rapp[1]

[1]Deutsches Zentrum für Luft- und Raumfahrt e.V. (DLR), Institut für Physik der Atmosphäre, 82234 Oberpfaffenhofen, Germany
[2]GATS, Boulder, CO, USA

**Correspondence:** Benjamin Witschas, (Benjamin.Witschas@dlr.de)

**Abstract.** In the course of the GW-LCYCLE II campaign, conducted in Jan/Feb 2016 from Kiruna, Sweden, coherent Doppler wind lidar (2-$\mu$m DWL) measurements were performed from the DLR Falcon aircraft to investigate gravity waves induced by flow across the Scandinavian Alps. During a mountain wave event on 28 January 2016, a novel momentum flux (MF) scan pattern with fore and aft propagating laser beams was applied to the 2-$\mu$m DWL. This allows to measure the vertical wind and the horizontal wind along the flight track simultaneously with a high horizontal resolution of $\approx 800$ m, and hence, enables to derive the horizontal MF profile for a broad wavelength spectrum from a few hundred meters to several hundred kilometers. The functionality of this method and the corresponding retrieval algorithm is validated using a comparison against in-situ wind data measured by the High Altitude and Long Range (HALO) aircraft which was also deployed in Kiruna for the POLSTRACC (Polar Stratosphere in a Changing Climate) campaign. Based on that, the systematic and random error of the wind speeds retrieved from the 2-$\mu$m DWL observations are determined. Further, the measurements performed on that day are used to reveal significant changes in the horizontal wavelengths of the vertical wind speed and of the leg-averaged momentum fluxes in the tropopause inversion layer (TIL) region, which are likely to be induced by interfacial waves as recently presented by Gisinger et al. (2020).

## 1 Introduction

Gravity waves (GWs) vertically connect the lower atmosphere, where they are primarily excited by flow over orography, convection, or flow deformation for instance caused by jets and fronts, with the upper atmosphere (Fritts and Alexander, 2003). While propagating, GWs carry momentum and energy and deposit them in regions where they dissipate. Whereas there is a general understanding of processes launching GWs, the nature of wave source spectra is more complex (Chen et al., 2007). Thus, there is still a need for a better characterization of GW sources to properly describe the dynamical coupling of the lower and the upper atmosphere. For this reason, several field campaigns with sophisticated airborne and ground-based instrumentation were performed within the last decades for instance the T-Rex campaign (Grubišić et al., 2008; Smith et al., 2008), the GW-LCYCLE I (Wagner et al., 2017; Ehard et al., 2016a) and GW-LCYCLE II campaign (Gisinger et al., 2020), DEEPWAVE (Fritts et al., 2016) and SOUTHTRAC (Rapp et al., 2020). During a few of these campaigns, it was demonstrated

that both ground-based and airborne lidar instruments are valuable for characterizing GW properties, as they provide vertically-
resolved information of dynamically relevant quantities for instance wind speed, temperature, and density. With such kind of
measurements, altitudes of $\approx 15$ to $100$ km are covered (e.g., Fritts et al., 2016; Kaifler et al., 2020; Kaifler and Kaifler, 2021;
Kaifler et al., 2021; Rapp et al., 2020).

To gain further knowledge about the excitation region of GWs in the troposphere, DLR's coherent Doppler wind lidar
($2$-$\mu$m DWL) was deployed on the Falcon aircraft during the GW-LCYCLE I (Gravity Wave Life Cycle) campaign (Dec 2013)
(Wagner et al., 2017; Ehard et al., 2016a) and the GW-LCYCLE II campaign (Jan/Feb 2016), both flown out of Kiruna, Sweden.
During GW-LCYCLE I, the $2$-$\mu$m DWL measured either horizontal or vertical wind speeds and demonstrated that the horizon-
tal scales of vertical wind and horizontal wind perturbations differ by an order of magnitude and that the spectral features can be
related to the high-frequency (vertical wind) and low-frequency (horizontal wind) part of the topography spectrum (Witschas
et al., 2017). A similar observation was reported by Smith and Kruse (2017) based on airborne in-situ measurements. Fur-
ther, Witschas et al. (2017) discussed the advantageousness of measuring the vertical wind speed $w$ and the horizontal wind
speed along flight direction $u_{\mathrm{par}}$ simultaneously with sufficient horizontal resolution of a few hundred meters, as this would
allow one to additionally quantify the propagation direction of the GWs and the corresponding momentum transport, especially
when the flight leg direction is chosen such that it matches the expected propagation direction of excited GWs. To do so, a
novel momentum flux (MF) scan pattern with fore and aft propagating laser beams was applied to the $2$-$\mu$m DWL for the first
time during the GW-LCYCLE II campaign. From that, the leg-averaged MF ($\mathrm{MF_x} = \overline{\rho}\,\overline{u'_{\mathrm{par}}\,w'}$) transported by GWs can be
derived (Eliassen and Palm, 1961; Smith et al., 2016), where the prime indicates the perturbations of the respective quantity,
the overline denotes the average along the flight leg and $\overline{\rho}$ denotes the mean air density.

In this paper, the novel MF-scan procedure and the corresponding retrieval algorithms are introduced employing $2$-$\mu$m DWL
measurements acquired on the flight performed on 28 January 2016 during a mountain wave (MW) event. The quality of
the derived wind components is estimated using a comparison against in-situ wind data measured by the High Altitude and
Long Range (HALO) aircraft which was flying coordinated with the Falcon aircraft on that day. Based on the $2$-$\mu$m DWL
data, significant changes in the horizontal wavelengths of the vertical wind speed and of the leg-averaged momentum fluxes
in the tropopause inversion layer (TIL) region are revealed. Whereas this paper concentrates on the description of the novel
measurement technique and the careful characterization of related uncertainties based on in-situ measurements and $2$-$\mu$m DWL
ground return analysis, the scientific results based on the retrieved leg-averaged MF profile have to some extend been published
by Gisinger et al. (2020), but are partly kept in this paper for the sake of completeness.

This paper is structured as follows. First, an overview of the GW-LCYCLE II campaign is given (Sect. 2), followed by a
short introduction of the $2$-$\mu$m DWL instrument (Sect. 3) including the description of the instrumental setup (Sect. 3.1), the
measurement principle (Sect. 3.2) as well as the data processing steps (Sect. 3.3). The results of the wind data analyses are
discussed in Sect. 4, followed by a summary given in Sect. 5.

## 2 The GW-LCYCLE II campaign

To investigate the appearance of GWs from their generation in the troposphere through to their dissipation, the GW-LCYCLE II (Gravity Wave Life Cycle Experiment II) campaign was conducted from 12 January 2016 to 3 February 2016 in Northern Scandinavia. With that, it extended the data set acquired during the precursor campaign GW-CYCLE I, which took place in December 2013 (Ehard et al., 2016b; Wagner et al., 2017; Witschas et al., 2017). Northern Scandinavia is a well-suited region to study coupling processes between the troposphere, the stratosphere, and the mesosphere, as the north-south orientation of the Scandinavian mountain ridge together with westerly blowing winds lead to the excitation of MWs that can then propagate upwards, provided that the background winds are favorable (Dörnbrack et al., 2001).

### 2.1 Instrumentation overview

During GW-LCYCLE II, several ground-based and airborne instruments at different sites were deployed. The Compact Rayleigh Autonomous Lidar - CORAL (Reichert et al., 2019; Kaifler and Kaifler, 2021), the Advanced Mesospheric Temperature Mapper - AMTM (Pautet et al., 2014) as well as the All-Sky Interferometric Meteor Radar - SKiYMET (Lukianova et al., 2018) were collocated in Sodankylä, Finland ($67.4°$ N, $26.6°$ E). The Middle Atmosphere Alomar Radar System - MAARSY (Latteck et al., 2012; Stober et al., 2012), a second AMTM, the Ground-based Infrared P-branch Spectrometer - GRIPS (Schmidt et al., 2013; Wüst et al., 2019), and the Alomar Rayleigh/Mie/Raman lidar system (Baumgarten, 2010) were operated on the west coast of Norway in Alomar ($69.3°$ N, $16.0°$ E). Another GRIPS instrument, an all-sky Fast Air-glow Imager - FAIM (Hannawald et al., 2016), a radiosonde launch-site as well as the DLR Falcon research aircraft were situated in Kiruna, Sweden ($67.8°$ N, $20.3°$ E). Whereas the ground-based instruments were mainly meant for probing the stratosphere and mesosphere, the airborne systems on board the Falcon were also used to probe the GW excitation region in the troposphere. Besides an airborne FAIM instrument (Wüst et al., 2019), and in-situ instruments used to measure tropospheric and stratospheric trace gases as water vapor concentration, $SO_2$, $CO$, $N_2O$, $CH_4$ and $CO_2$, the Falcon was equipped with a downward looking coherent $2$-$\mu$m DWL that was demonstrated to be useful for characterizing the spectral features of MWs by exploiting measurements of either horizontal or vertical wind speed profiles, respectively (Witschas et al., 2017). In addition to the $2$-$\mu$m DWL wind data, the horizontal and the vertical wind speed were measured at flight level by the Falcon nose-boom employing a flow angle sensor (Rosemount model 858) together with an inertial reference system (Honeywell Lasernav YG 1779) as described by Bögel and Baumann (1991) and Krautstrunk and Giez (2012). Jointly with the Falcon, the HALO aircraft was deployed in Kiruna, Sweden in the same period, aiming to investigate the Polar Stratosphere in a Changing Climate (POLSTRACC). For that purpose, HALO was equipped with 10 in-situ and 3 remote sensing instruments measuring the composition of the upper troposphere and the lower stratosphere. A detailed overview of the POLSTRACC campaign, the scientific goals, and the instrumentation is given by Oelhaf et al. (2019).

A summary of the Falcon flight tracks performed during GW-LCYCLE II is shown in Fig. 1. Altogether, 6 research flights with a total of about 20 flight hours were performed. The flights were planned along and across the Scandinavian Alps to probe the excitation region of MWs. The HALO flight track flown on 28 January 2016 is additionally plotted (yellow line) as

it was particularly coordinated with the two Falcon flights performed on that day (dark and light blue lines). The geolocation of Kiruna airport is indicated by the black cross, and the flight leg (FL3), on which the novel MF-scan was applied to the 2-$\mu$m DWL is marked by the black dots.

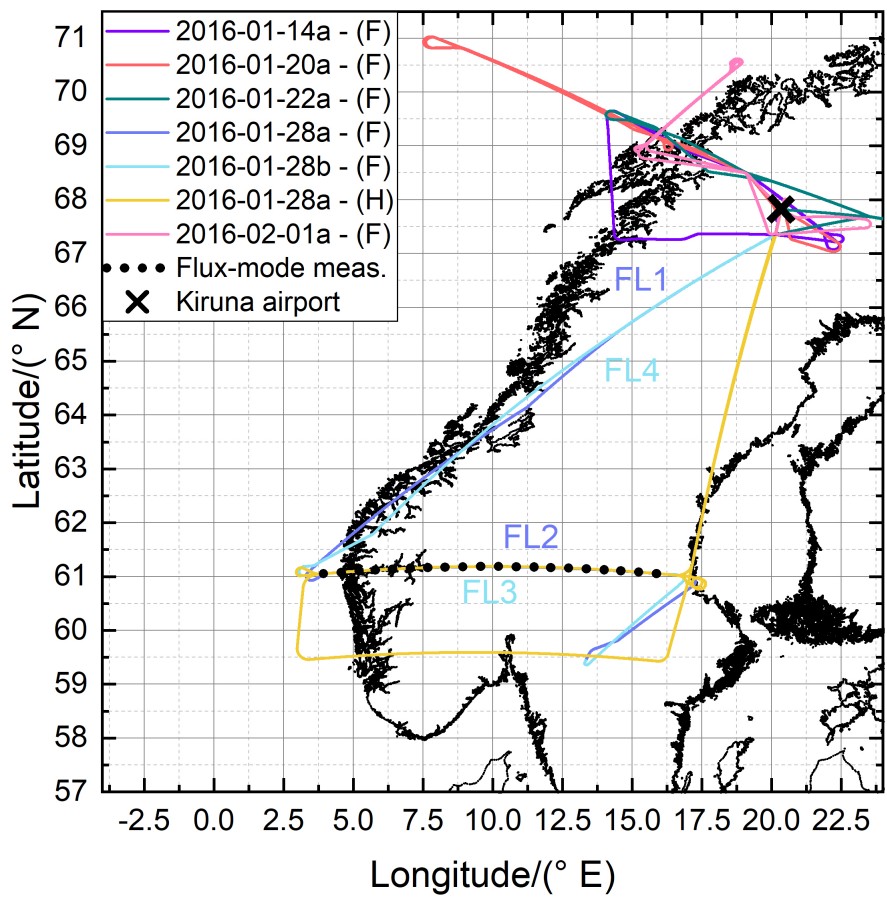

**Figure 1.** Overview of the Falcon (F) flight tracks flown during GW-LCYCLE II and the coordinated HALO (H) flight track flown on 28 January 2016 (see label). The black dots indicate the region where the MF mode scan pattern was performed. The black cross denotes the geolocation of Kiruna airport. An overview of the respective 2-$\mu$m DWL measurement modes that were applied during the respective flight legs (FL) flown on 28 January 2016 is given in table 1.

## 2.2 Coordinated research flights on 28 January 2016

The 28 January 2016 was a favorable day because of several reasons. First, the excitation of MWs was likely due to moderate wind speeds blowing perpendicular to the Scandinavian mountain ridge. Second, the research flights of HALO and Falcon were coordinated, giving the possibility of using additional measurement data for GW characterization. And third, the 2-$\mu$m DWL

was operated with a new MF scanning mode (MF-mode) to measure the vertical and the horizontal wind along the flight track, simultaneously with a high horizontal resolution of a few hundred meters.

The meteorological situation on 28 January 2016 was featured by two synoptic low-pressure systems, one over the tip of Greenland and one over the Baltic sea, leading to moderate south-westerly winds of $10 \, \mathrm{m \, s^{-1}}$ to $20 \, \mathrm{m \, s^{-1}}$ in the troposphere and thus, to the excitation of MWs over the Scandinavian mountain range (Gisinger et al., 2020). For that reason, a coordinated flight of Falcon and HALO was planned and conducted on this day (Fig. 1, in dark and light blue (Falcon) and yellow (HALO)). The Falcon took off from Kiruna airport at 12:45 UTC and climbed to an altitude of about 9.8 km. The first flight leg (Fig. 1, FL1) started in southern direction along the Scandinavian coast with the 2-$\mu$m DWL operating in wind-mode to derive both, wind speed and wind direction with a horizontal resolution of $\approx 8.4$ km and a vertical resolution of 100 m (see also Sect. 3.2), hence, giving the possibility to investigate the inflow conditions. At about $61°$ N, the Falcon turned eastwards and flew a 700 km long cross-mountain leg (Fig. 1, FL2) before doing a refueling-stopover at Karlstad airport, Sweden. On this leg, the 2-$\mu$m DWL performed in vertical-wind-mode where the laser beam is pointed to nadir-direction to measure the vertical wind speed with a horizontal resolution of $\approx 200$ m and a vertical resolution of 100 m (see also Sect. 3.2), and hence, enables to resolve the small-scale structure of the excited MWs. At 17:10, the Falcon lifted off again, and flew the similar flight track back. On the cross-mountain leg (Fig. 1, FL3), the 2-$\mu$m DWL was operated in MF-mode to determine the leg-averaged MF profile. At about $61°$ N, the Falcon turned northwards and flew back to Kiruna airport (Fig. 1, FL4), while measuring with the 2-$\mu$m DWL in wind-mode again to verify if the inflow conditions have changed since the first flight, which was performed about 4 hours earlier. An overview of the four different flight legs including the start and stop latitudes and longitudes, the leg length and the applied 2-$\mu$m DWL scanning mode, is given in table 1. The respective characteristics of the 2-$\mu$m DWL measurement modes, namely wind-mode, vertical-wind-mode, and MF-mode, are presented in detail in Sect. 3.2.

**Table 1.** Overview Falcon flight legs performed on 28 January 2016 including the corresponding 2-$\mu$m DWL measurement modes and related data products.

| Nr. | Flight | Time/(UTC)[*] | Lat./($°$ N)[*] | Lon./($°$ E)[*] | Length/(km) | 2-$\mu$m DWL mode | data product |
|-----|--------|----------|----------|----------|-------------|------------------|--------------|
| FL1 | 20160128a | 12:56/14:27 | 67.19/61.21 | 19.57/3.65 | 1019 | Wind-mode | 3D Wind vector |
| FL2 | 20160128a | 14:33/15:23 | 61.02/61.02 | 3.89/16.76 | 703 | Vertical-wind-mode | Vertical wind speed |
| FL3 | 20160128b | 17:38/18:46 | 61.04/61.02 | 16.23/3.23 | 713 | MF-mode | Vert./Hor. wind[†] |
| FL4 | 20160128b | 18:48/20:02 | 61.17/66.94 | 3.14/18.73 | 1002 | Wind-mode | 3D Wind vector |

[*] Values are start/stop values of the corresponding flight leg (LAT = latitude, LON = longitude).

[†] Horizontal wind is measured along the flight direction ($u_{\mathrm{par}}$).

HALO left Kiruna airport at 16:16 UTC, and thus, a little later than originally planned due to a technical problem with one of the instruments aboard. Still, it was possible to perform coordinated measurements with the Falcon aircraft. In particular, while the Falcon was flying the cross-mountain leg FL3 at an altitude of 9.8 km, HALO flew below at an altitude of 7.8 km. The horizontal distance between both aircraft (in latitude-direction) was less than 100 m, and the temporal distance was varying between 1 and 2 minutes. Hence, the HALO in-situ measurements of wind speed and direction represent a perfect

validation data set for the MF-mode measurements and the corresponding wind retrieval algorithm as discussed in Section 3.3 and Section 4.2.

## 3  The 2-$\mu$m Doppler Wind Lidar at DLR

The DLR 2-$\mu$m DWL has been deployed in several airborne campaigns within the last two decades to measure for instance the optical properties of aerosols (Chouza et al., 2015, 2017) and horizontal wind speeds over the Atlantic Ocean as input data for numerical weather prediction assimilation experiments (Weissmann et al., 2005; Schäfler et al., 2018). Furthermore, it was extensively used for pre-launch (Marksteiner et al., 2018; Lux et al., 2018) and post-launch validation activities of the first space-borne wind lidar Aeolus (Witschas et al., 2020; Lux et al., 2020; Witschas et al., 2022; Lux et al., 2022). Additionally, horizontal and vertical wind speed measurements have been performed and have been used to characterize orographically induced gravity waves (Witschas et al., 2017; Wagner et al., 2017; Bramberger et al., 2017; Gisinger et al., 2020).

In this section, the 2-$\mu$m DWL instrumental setup is shortly described (section 3.1), followed by an explanation of the corresponding measurement principle and the applied scanning modes in section 3.2. Afterwards, the data processing steps are discussed in section 3.3, concentrating on the novel MF retrieval. Details about the retrieval procedures of the vertical as well as the horizontal wind speed and direction are given by Witschas et al. (2017).

### 3.1  Instrumental description

A picture of the 2-$\mu$m DWL mounted within the Falcon aircraft is shown in Fig. 2. The 2-$\mu$m DWL is a heterodyne-detection wind lidar composed of several sub-units. The heart of the system is the transceiver unit which consists of the laser transmitter, an 11 cm diameter afocal telescope, receiver optics, and a double wedge scanner (Fig. 2, (1)). The transceiver unit (excluding the scanner) was built by CLR Photonics (Henderson et al., 1991, 1993; Hannon and Henderson, 1995) and delivered to DLR in 1999. The double-wedge scanner and the data acquisition unit were developed and constructed by DLR (Witschas et al., 2017). The power supply and cooling unit of the system are mounted in a rack located beside the transceiver (Fig. 2, (2)). The corresponding control electronics and the data acquisition unit are mounted in a second rack (Fig. 2, (3)).

The 2-$\mu$m DWL system is based on a Tm:LuAG laser producing laser pulses with a wavelength of 2022.54 nm (vacuum), 1 mJ to 2 mJ energy, and a repetition rate of 500 Hz, ensuring eye-safe operation. Before being sent to the atmosphere, the laser beam is expanded by the telescope to a diameter of $\approx 10$ mm. The small portion of the light that is scattered back to the instrument is collected with the same telescope and afterwards directed to the signal detector, where it is mixed with a local oscillator laser that is also used for injection seeding of the outgoing laser pulse. The time-resolved detector signal resulting from each single laser shot is sampled with 500 MHz and 8 bit resolution. The availability of the signal on a single-shot basis gives maximum flexibility for post-processing (see also section 3.3), as for instance the removal of the signal from unseeded laser pulses and the correction of potential laser frequency variations that may occur, especially within the harsh environment of an aircraft. To do so, the beat signal of the emitted laser pulse and the local oscillator (seed laser) is analyzed. When the beat frequency or the laser pulse built-up time exceeds a certain difference from their nominal values, the signal from the

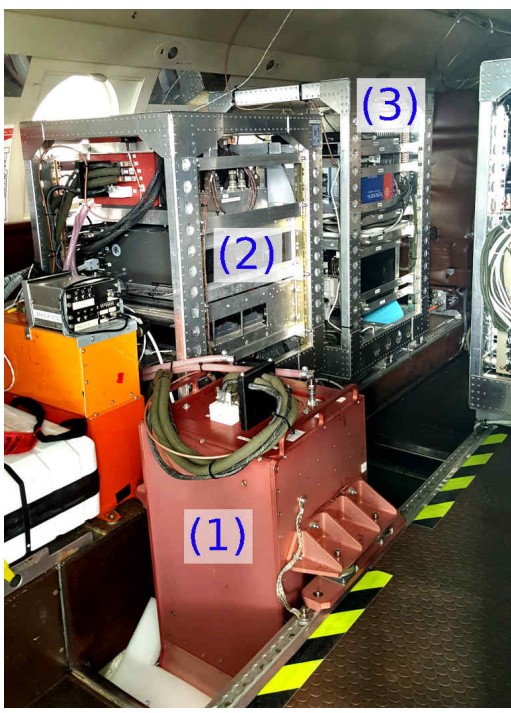

**Figure 2.** Picture of the 2-$\mu$m DWL mounted within the Falcon aircraft showing the transceiver unit (1), the power supply and cooling rack (2) as well as the data acquisition rack (3).

respective laser pulse is not considered for accumulation. Moreover, before accumulating the reference pulse spectra, they are frequency-shifted to a defined reference value to correct for pulse-to-pulse frequency variations, and thus, to avoid spectral

broadening in the accumulation process. A similar frequency shift is then also applied to the atmospheric signal power spectra. Afterwards, the signals of the remaining laser pulses in a one-second time interval are spectrally averaged. Beforehand, the detector raw signal of the atmospheric return was divided into segments that lead to 100-m range gates in the vertical by considering the actual laser beam pointing angle, the aircraft altitude and attitude, and the reference pulse timing. The detector signal at the end of the record (below the ground return) is used to analyze and correct the detector noise characteristics. The

power spectrum of the noise signal is calculated for each single laser pulse and is additionally averaged for a one-second time interval. Consequently, the power spectrum for each single range gate is divided by the respective noise spectrum to correct for the system noise and the receiver frequency response (noise whitening). In the next step, the power spectra are corrected for the aircraft speed projected onto the LOS direction, which is derived from the ground speed measured by the GPS module and the actual laser beam pointing direction. The latter is calculated from the actual scanner position, the aircraft attitude (inertial

reference system) as well as the lidar installation position with respect to the aircraft reference frame (Chouza et al., 2016). Finally, the LOS wind speed $v_{\mathrm{LOS}}$ is calculated according to $v_{\mathrm{LOS}} = (\Delta f c)/(2 f_0)$, where $\Delta f$ is the frequency shift between the reference pulse and the atmospheric signal, $f_0$ is the laser frequency, $c$ is the velocity of light, and $\lambda_0 = c/f_0 = 2022.54$ nm

is the laser wavelength. To retrieve vertical or horizontal wind information from respective LOS wind measurements, further processing steps are needed, as described in Sect. 3.2.

## 3.2 Measurement principle

Commonly, the 2-$\mu$m DWL is used to either measure the three-dimensional wind vector (wind-mode) or rather to measure the vertical wind speed (vertical-wind-mode). When operating in wind-mode, the velocity-azimuth display (VAD) scan technique is applied (Browning and Wexler, 1968) by performing a conical scan around the vertical axis with an off-nadir angle of $20°$. Typically, one scanner revolution with 21 line-of-sight (LOS) measurements separated by $18°$ in the azimuth direction takes about $42$ s. By further considering the aircraft speed of about $200$ m s$^{-1}$, the horizontal resolution of wind-mode observations is about $8.4$ km, depending on the actual ground speed of the aircraft. Hence, the vertical velocity wind field, which is shaped by scales of tens of kilometers and smaller in the presence of MWs (Smith and Kruse, 2017), is not well resolved in wind-mode data. The vertical resolution is set by the spatial averaging interval to be $100$ m, taking into account the full width at half maximum of the laser pulse of $\approx 400$ ns.

When measuring in vertical-wind-mode, the laser beam is pointed to nadir-direction, and thus, the measured LOS wind equals the vertical wind speed. To avoid an additional contribution of the horizontal wind speed, the aircraft attitude variations (pitch and roll angles) during cruising flight are compensated by a scanner control loop on a 1-second basis. As one LOS measurement is averaged over $1$ s (500 laser pulses), the horizontal resolution of the vertical wind speed data is about $200$ m and the vertical resolution is $100$ m as for the wind-mode measurements.

Based on 2-$\mu$m DWL data acquired during GW-LCYCLE I, it was revealed that both wind-mode and vertical-wind-mode profiles are beneficial for characterizing the spectral properties of GWs and their evolution while they are propagating (Witschas et al., 2017). However, it was also discussed that simultaneous measurements of the horizontal and the vertical wind speed with a horizontal resolution of a few hundred meters and a low representativeness error would be even more beneficial, as such measurements would allow retrieving the vertical flux of horizontal momentum induced by GWs (Smith et al., 2008, 2016). To cope with that issue, a new scan pattern with alternately fore and aft propagating laser beams (MF-mode) was applied to the 2-$\mu$m DWL for the first time during the GW-LCYCLE II campaign (see also Sect. 3.3 and Fig. 3). A similar approach was already introduced by Vincent and Reid (1983) using a ground-based Doppler radar. Here, this approach is adapted to airborne wind lidar measurements, which additionally enable spatial discrimination compared to the ground-based radar measurements. Considering that the wind field is constant for the time of intersecting fore and aft laser beam pairs, this scan pattern gives the possibility to retrieve both vertical wind speed and the horizontal wind component along the flight direction. The assumption of a constant wind field is justifiable, considering that the duration between intersecting beams is about $36$ s on ground and respectively less for altitudes closer to the aircraft for the actual measurement configuration (off-nadir angle of $20°$, flight altitude of $9.8$ km, aircraft speed of about $200$ m s$^{-1}$, and LOS averaging time of $2$ s). Due to the rather clear air conditions during the flight, it was decided to keep each LOS pointing direction for $2$ s, instead of the usually used $1$ s time interval, to assure reasonable data coverage. This means that the retrieved MF-scan data ($u_{\mathrm{par}}$ and $w$) provides a horizontal resolution of $\approx 800$ m and a vertical resolution of $100$ m, which is high enough to resolve even small-scale MWs with horizontal wavelengths

of 1.6 km and larger. For future applications, it is foreseen to reduce the time interval of each pointing direction to 1 s halving the horizontal resolution to $\approx 400$ m.

## 3.3 Data processing

When operating in MF-mode, the 2-$\mu$m DWL scanner steers the laser beam alternately fore (Fig. 3, blue) and aft (Fig. 3, green) with a certain off-nadir angle $\theta$ (usually $\pm 20°$) as illustrated in Fig. 3. This procedure results in LOS wind measurements in forward direction $v_{f_i}$, and in backward direction $v_{b_i}$ according to

$$v_{f_i}(R, \theta_{f_i}, x) = w(R, \theta_{f_i}, x)\cos(\theta_{f_i}) + u_{par}(R, \theta_{f_i}, x)\sin(\theta_{f_i})$$
$$v_{b_i}(R, \theta_{b_i}, x) = w(R, \theta_{b_i}, x)\cos(\theta_{b_i}) - u_{par}(R, \theta_{b_i}, x)\sin(\theta_{b_i}) \tag{1}$$

where $\theta_{f_i} \approx 20°$ and $\theta_{b_i} \approx -20°$ are the actual off-nadir angles of the forward and backward directed laser beam, respectively, $x$ denotes the horizontal distance and $R$ the vertical distance from the aircraft. As the laser beam pointing direction, and thus, the off-nadir angles are accurately known from the scanner position and the aircraft attitude measured by an inertial reference system, the two remaining unknowns, namely the horizontal wind speed along flight direction $u_{par}$ and the vertical wind speed $w$, can directly be derived from a successive pair of $v_{f_i}, v_{b_i}$ measurements according to

$$u_{par}(x, R) = \csc(\theta_{f_i} - \theta_{b_i})(v_{f_i}\cos\theta_{b_i} - v_{b_i}\cos\theta_{f_i})$$
$$w(x, R) = \csc(\theta_{f_i} - \theta_{b_i})(-v_{f_i}\sin\theta_{b_i} + v_{b_i}\sin\theta_{f_i}) \tag{2}$$

where $\csc(z) = 1/\sin(z)$ denotes the cosecant function. The beam pairs have to be chosen such that they intersect at the respective horizontal position $x$ and vertical position $R$ as illustrated in Fig. 3. To obtain a continuous data set for the fore and aft measured LOS velocities, the data is linearly interpolated.

## 4 Results

In this section, the findings obtained from the 2-$\mu$m DWL measurements acquired on 28 January 2016 are discussed. In Sect. 4.1, the results from horizontal and vertical wind observations are shown, revealing the overall inflow conditions as well as pronounced changes of the horizontal scales in the vertical velocity field. In Sect. 4.2, the accuracy and precision of the MF-mode measurements are estimated by a comparison against HALO in-situ wind measurements. In Sect. 4.3, the characteristics of the derived leg-averaged horizontal MF profile are discussed.

### 4.1 Horizontal and vertical wind speed

The horizontal wind speed and wind direction measured by the 2-$\mu$m DWL on leg FL1 and leg FL4 are shown in Fig. 4. It can be seen that the horizontal wind was blowing with up to $35$ m s$^{-1}$ at altitudes between $4$ km and $10$ km from westerly directions, and that the largest wind speeds were located between $62°$ N and $63°$ N. As this situation was forecasted well, the cross-mountain leg was flown south of Kiruna ($\approx 61°$ N) to probe the GW excitation region. At this latitude, the wind

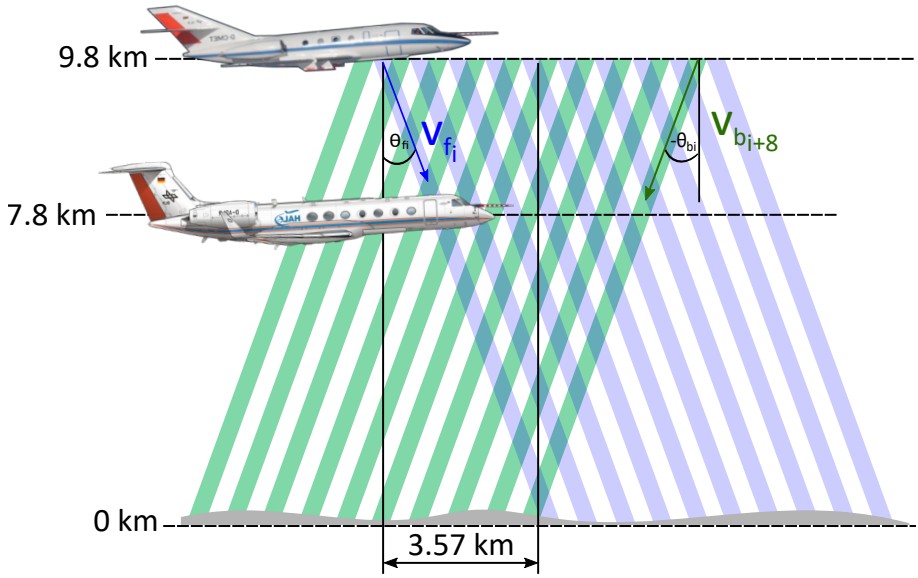

**Figure 3.** Schematic illustration of the scan procedure applied when operating in MF-mode to retrieve profiles of the horizontal wind along flight direction $u_{\mathrm{par}}$ and the vertical wind $w$, simultaneously.

direction was measured to be between $260°$ N and $300°$ N, and thus, almost perpendicular to the Scandinavian mountain ridge
and along the Falcon flight track direction. Thus, this situation provided excellent conditions for both, the excitation of MWs
as well as for the first application of MF-mode. By comparing the data from FL1 (Fig.4, a, c) and FL4 (Fig.4, b, d), it can be
furthermore realized that the horizontal wind speed slightly increased to up to $40\,\mathrm{m\,s^{-1}}$, and that the location of the local wind
speed maximum moved north. Still, the wind direction remained perpendicular to the mountain ridge for all altitudes, which
confirms the assumption of steady inflow conditions during the observation period.

At about $61°$ N, the Falcon turned eastward and performed the first cross-mountain leg, while the 2-$\mu$m DWL was measuring
in vertical-wind-mode. The corresponding data is shown in Fig. 5. It is obvious that the data coverage of the lidar measurements
is sparse, which is on the one hand due to the aerosol-poor atmospheric conditions, and on the other hand due to clouds at the
tropopause level and below ($7\,\mathrm{km}$ to $8\,\mathrm{km}$), which prevents the laser beam to propagate to lower altitudes. Still, from the data it
can be seen that MWs are excited. The vertical wind speed along the flight leg varies between $\pm 3\,\mathrm{m\,s^{-1}}$ and the phase lines are
vertically orientated. In the western part of the flight leg, the wind speeds are smaller ($\pm 1\,\mathrm{m\,s^{-1}}$), and get larger while going
eastward (downstream) to the lee side of the mountains. Analyzing the spectral structure of the excited GWs, an interesting
behavior can be observed between $5°$ E to $8°$ E (see also the light-gray line in Fig. 5). Below altitudes of $\approx 8.5\,\mathrm{km}$, and thus
below the thermal tropopause, which is determined using European Centre for Medium-Range Weather Forecasts (ECMWF)
model data, the waves have a horizontal wavelength of about $20\,\mathrm{km}$, whereas the ones in or rather above the tropopause have
a horizontal wavelength of only $10\,\mathrm{km}$ and smaller. As shown by Gisinger et al. (2020), this behavior can be explained by the
occurrence of interfacial waves with wavelengths smaller than $10\,\mathrm{km}$, that are induced by partial reflection of longer waves at

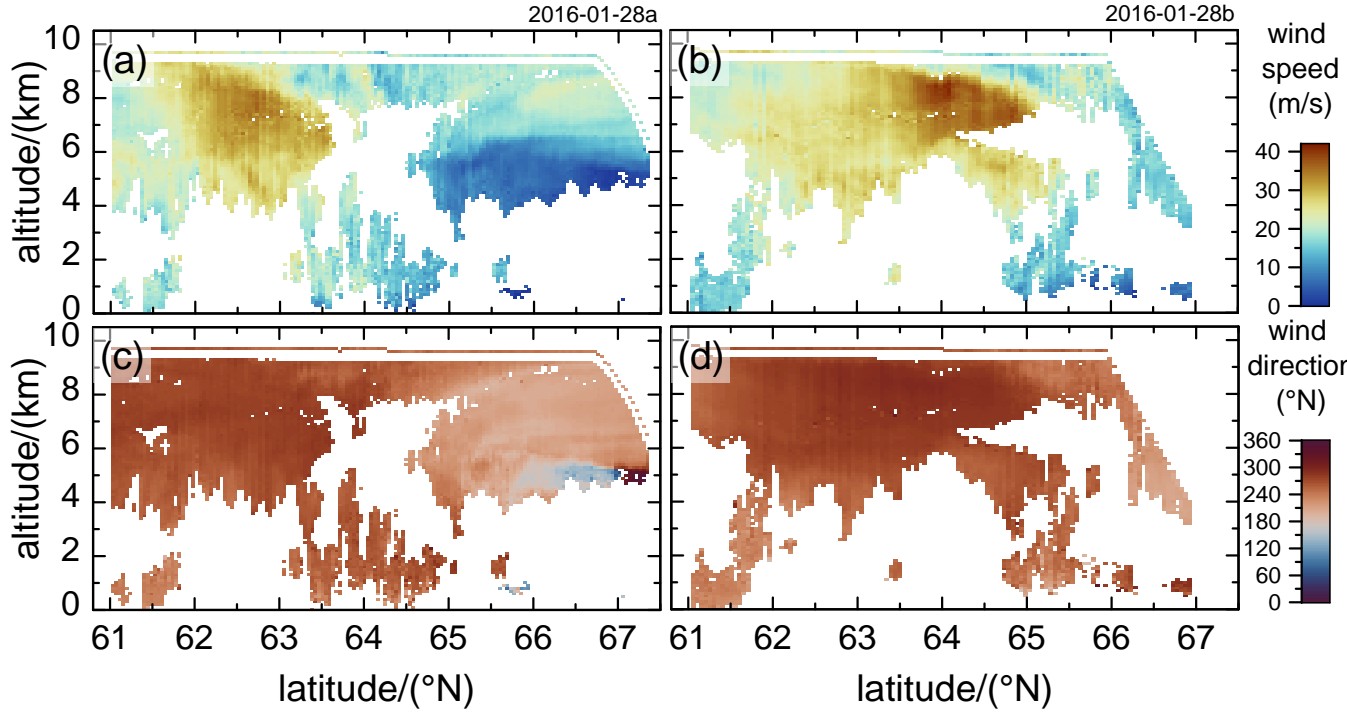

**Figure 4.** Horizontal wind speed (top, panel a and b) and wind direction (bottom, panel c and d) derived from 2-$\mu$m DWL observations acquired in wind-mode on the flight legs FL1 (left) and FL4 (right) flown on 28 January 2016 (see also table 1, and Fig. 1). The corresponding wind speed and direction measured in-situ by the Falcon nose-boom is indicated by the thin line at flight level.

the TIL region. Currently, further numerical simulations are ongoing to investigate if the generation and downward propagation of secondary waves in the stratosphere also contribute to the observed phenomenon. Further eastward (8° E to 16° E), only the MWs with the shorter wavelengths ($\lambda < 10$ km) are visible as clouds at the tropopause prevented measurements below. At the end of the flight leg (16° E to 17° E), the spectral change of the MWs at the TIL region (at $\approx 8$ km altitude) is again obvious. This result demonstrates that the height-resolved vertical-wind-mode data is useful to observe dynamical changes in the wave field with altitude.

### 4.2 Accuracy of MF-mode measurements

During several campaigns within the last years, the accuracy (systematic error) and the precision (random error) of 2-$\mu$m DWL wind-mode and vertical-wind-mode observations were determined utilizing dropsonde comparisons. In particular, the random error of single LOS measurements is characterized to be $0.2$ m s$^{-1}$ and the systematic error is smaller than $0.05$ m s$^{-1}$ (Witschas et al., 2017). The mean errors for wind-mode observations varies between $-0.03$ m s$^{-1}$ and $0.08$ m s$^{-1}$ (systematic error) and $0.92$ m s$^{-1}$ and $1.50$ m s$^{-1}$ (random error), whereas these errors contain contributions from both, the 2-$\mu$m DWL

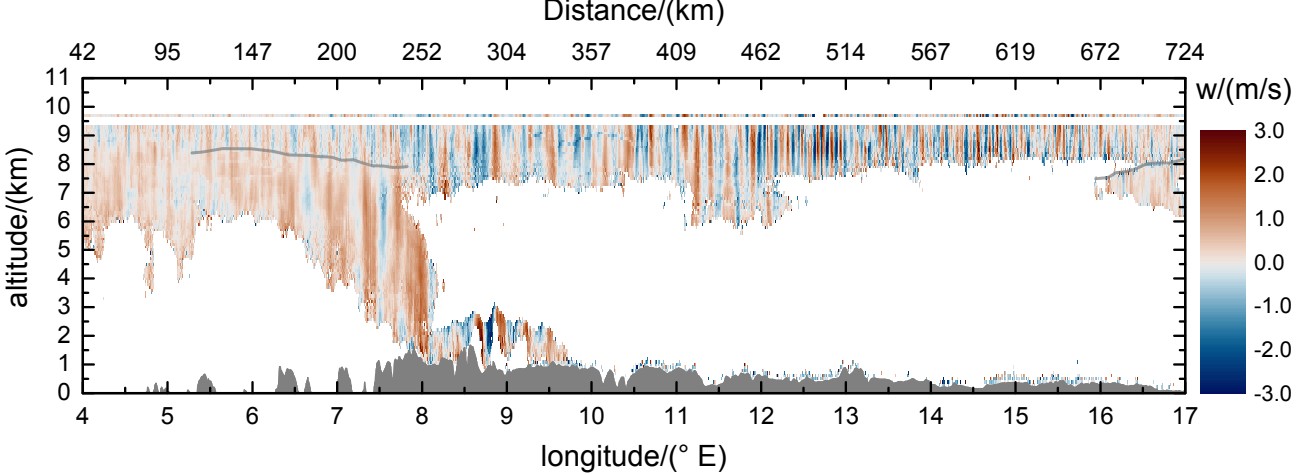

**Figure 5.** Vertical wind speed derived from 2-$\mu$m DWL observations acquired in vertical-wind-mode on the flight leg FL2 flown on 28 January 2016 (see also table 1, and Fig. 1). The corresponding vertical wind measured in-situ by the Falcon nose-boom is indicated by the thin line at flight level. The surface elevation is sketched by the gray area at the bottom of the graph. The light-gray line between 7.5 km and 8.5 km altitude indicates the position of the TIL, where the spectral properties of the GWs can be seen to change remarkably.

and the dropsondes (Witschas et al., 2020; Weissmann et al., 2005; Chouza et al., 2016; Reitebuch et al., 2017; Schäfler et al., 2018; Witschas et al., 2017).

As the temporal and spatial resolution, as well as the representativity, is different for MF-mode measurements, it was questionable how this affects the systematic and random error of the retrieved wind data. For that reason, HALO in-situ wind data, acquired on the coordinated flight leg with Falcon and HALO on 28 January 2016 (Fig. 1, FL3) as well as ground return analyses are used for validation. During this flight leg, the Falcon aircraft was flying at 9.8 km altitude, whereas HALO was flying at 7.8 km altitude. The systematic and random errors of HALO in-situ wind measurements are estimated to be 0.4 m s$^{-1}$ and 0.25 m s$^{-1}$ for the vertical wind speed or rather 0.6 m s$^{-1}$ and 0.3 m s$^{-1}$ for the horizontal wind speed (Mallaun et al., 2015; Giez et al., 2017, 2021), whereas the similar uncertainty is considered for the horizontal wind along flight direction ($u_{\mathrm{par}}$).

The vertical and the horizontal wind speed along flight direction retrieved from the 2-$\mu$m DWL data at 7.8 km are shown in Fig. 6 by the black line in panel (a) and (b), respectively. The corresponding data measured in-situ on the HALO aircraft is indicated by the red line. The bottom panels (c) and (d) represent the corresponding residuals. It can be seen that both data sets are in great accordance along flight leg with its length of almost 700 km. The mean of the vertical wind speed residual data, which is an estimate of the mean systematic error, is 0.01 m s$^{-1}$, and the corresponding standard deviation is 0.3 m s$^{-1}$. This demonstrates that the vertical wind speed retrieved from the MF-mode data is almost bias-free, and the random error $\sigma_{\mathrm{DWL_{vert}}}$ is $\approx 0.17$ m s$^{-1}$, considering the specified random error of HALO vertical wind speed measurements of 0.25 m s$^{-1}$ and assuming the errors to be uncorrelated ($\sigma_{\mathrm{DWL_{vert}}} = ((0.3 \text{ m s}^{-1})^2 - (0.25 \text{ m s}^{-1})^2)^{1/2} = 0.17 \text{ m s}^{-1}$). Furthermore, it can be seen, that small discrepancies of up to 0.5 m s$^{-1}$ originate in the region between 14° E and 15.5° E. Investigations of the HALO attitude revealed, that the ground speed in this region changed from about 180 m s$^{-1}$ to 207 m s$^{-1}$ resulting in a pitch

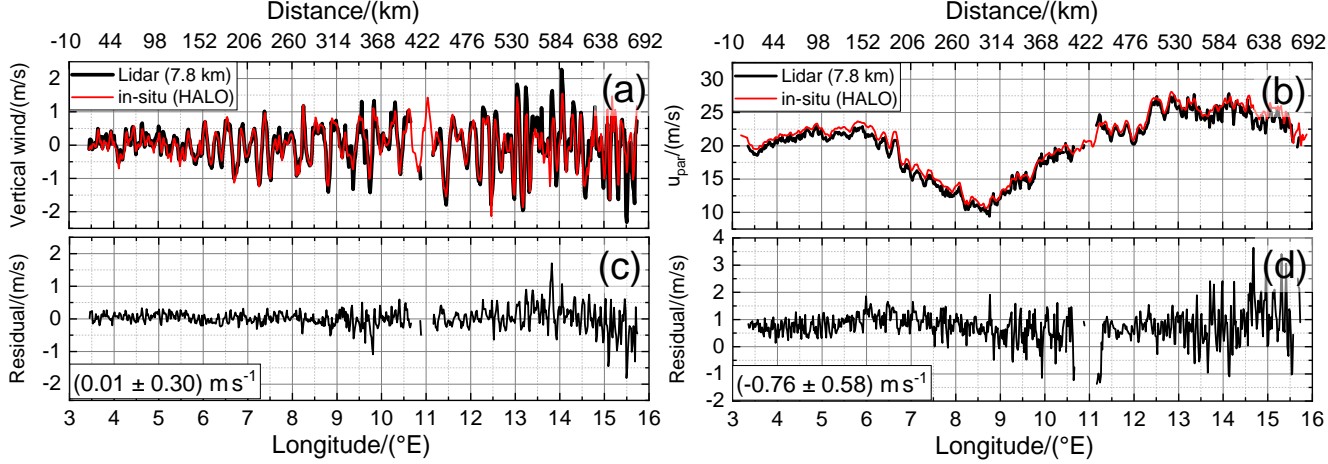

**Figure 6.** Vertical wind speed (a) and horizontal wind speed along flight direction (b) derived from 2-$\mu$m DWL observations acquired in MF-mode on FL3 at 7.8 km altitude (black), and the wind speed measured in-situ at the HALO aircraft flying at the same altitude (red). The bottom panels (c) and (d) indicate the corresponding residuals.

angle change from $\approx 1.4°$ to $\approx 2.0°$, whereas these changes might lead to a systematic error in the retrieved in-situ winds. On the other hand, as discussed later in Fig. 7, the 2-$\mu$m DWL ground returns do not show noticeable deviations in this region,
further suggesting that the lidar measurements are reliable.

For $u_{\mathrm{par}}$, the mean of the residual data is $-0.76$ m s$^{-1}$ and the corresponding standard deviation is $0.58$ m s$^{-1}$. The slightly larger error is probably due to slightly different heading angles of the two aircraft flying at different altitudes which may lead to uncertainties in the wind projection due to the slightly different wind directions retrieved from the respective measurements. Moreover, the 2-$\mu$m DWL data represents the mean wind speed at an altitude of $(7.8 \pm 0.05)$ km, whereas the HALO in-situ
data represents the wind speed at flight level. Thus, vertical gradients in the horizontal wind field could also lead to the observed difference in $u_{\mathrm{par}}$ between 2-$\mu$m DWL and HALO data. Still, all GW induced structures are visible in both datasets. And, as the MF calculation is based on the perturbations of $u_{\mathrm{par}}$, the enhanced systematic error is negligible in this particular case. Considering the random error of $0.3$ m s$^{-1}$ specified for HALO horizontal wind speed measurements, the random error $\sigma_{\mathrm{DWL_{hor}}}$ of $u_{\mathrm{par}}$ can be estimated to be $0.5$ m s$^{-1}$ or better ($\sigma_{\mathrm{DWL_{hor}}} = ((0.58 \text{ m s}^{-1})^2 - (0.30 \text{ m s}^{-1})^2)^{1/2} = 0.50$ m s$^{-1}$). As
for the vertical wind speed, slightly larger discrepancies are obvious in the region from $14°$ E to $15.5°$ E, which are likely to be caused by the aforementioned aircraft attitude changes and corresponding impacts on the in-situ measurements.

In contrast to the in-situ measurements, airborne wind lidar measurements enable to estimate the accuracy of the performed wind observations by analyzing the returns from the non-moving ground, which should yield values of $0$ m s$^{-1}$ by definition.
For the discussed flight leg, ground returns are available from about $9.5°$ E to $15.8°$ E as shown in Fig. 7, black. Further west, clouds prevented the laser beam to hint the ground. A moving average of 10 successive data points is additionally indicated

by the orange line. It can be seen that the LOS wind speeds are varying around $0~\mathrm{m~s^{-1}}$ with peak-to-peak values mostly below $0.1~\mathrm{m~s^{-1}}$. The mean of this data set yields a value of $0.00~\mathrm{m~s^{-1}}$, and a standard deviation of $0.05~\mathrm{m~s^{-1}}$, further demonstrating the high accuracy of MF-mode wind data acquired on this flight leg. Furthermore, flight attitude changes that occurred also for the Falcon between $14°$ E to $15.5°$ E induce a systematic deviation of less than $-0.05~\mathrm{m~s^{-1}}$, confirming that the discrepancy of 2-$\mu$m DWL and HALO in-situ data is unlikely to be caused by the lidar measurements.

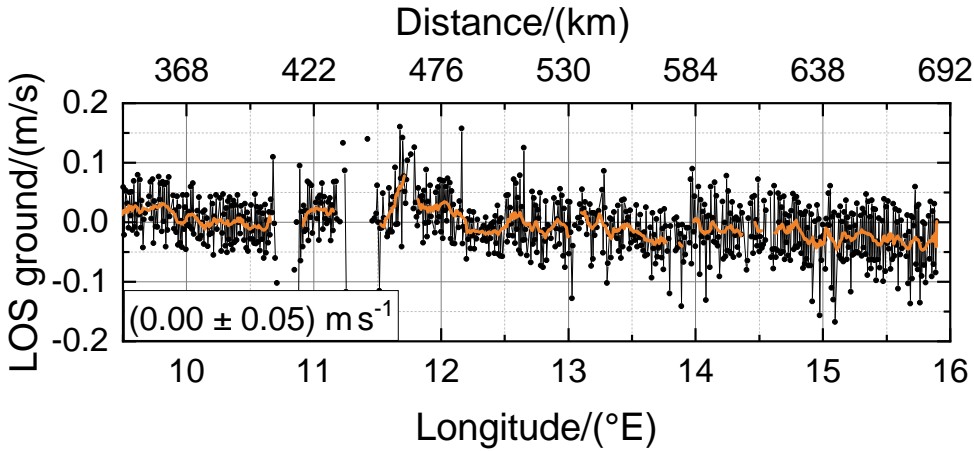

**Figure 7.** LOS wind speed retrieved from 2-$\mu$m DWL ground returns (black) and moving average of 10 successive data points (orange).

### 4.3 Momentum flux profile

From the simultaneous measurements of $u_{\mathrm{par}}$, and $w$, the spatial leg-averaged vertical flux of horizontal momentum $\mathrm{MF_x}$ can be derived as for instance introduced by Smith et al. (2016) according to

$$\mathrm{MF_x}(z) = \overline{\rho}(z)\,\overline{u'_{\mathrm{par}}\,w'}, \tag{3}$$

where $\overline{\rho}$ is the mean air density taken from ECMWF integrated forecast system (IFS) data at respective altitudes $z$ along the flight leg, which is usually several hundred kilometers long, and thus, usually longer than the expected GW wavelength, depending on the spectral filters that are applied. The quantities $u'_{\mathrm{par}}$ and $w'$ are the perturbations of the horizontal wind speed along the flight track and the vertical wind speed, respectively, and are derived by applying a Butterworth filter with different cut-off frequencies (Ehard et al., 2015). In particular, $u'_{\mathrm{par}}$ and $w'$ are determined for two different wave classes namely short waves which cover all wavelengths of smaller than $30~\mathrm{km}$, and long waves that cover wavelengths of larger than $30~\mathrm{km}$.

The uncertainty of the derived momentum fluxes is estimated as suggested by Smith et al. (2016) based on the random errors of $u_{\mathrm{par}}$ and $w$. To propagate these errors for the MF derivation, a transect with anti-correlated sinusoidal $u'_{\mathrm{par}}$ and $w'$ oscillations is considered with common amplitudes of $5.0~\mathrm{m~s^{-1}}$ and $1.0~\mathrm{m~s^{-1}}$, respectively. Furthermore, a mean air density of $\overline{\rho} = 0.5~\mathrm{kg~m^{-3}}$ for the altitude range between $7.0~\mathrm{km}$ and $10.0~\mathrm{km}$ is assumed. This enables to define a reference value

of the leg-averaged momentum flux $\mathrm{MF_x} = \bar{\rho}\,\overline{u'_{\mathrm{par}}\,w'} = (0.5)(0.5)(5.0)(1.0) = 1.25$ Pa. In a worst-case situation, the random errors $\sigma_{u'_{\mathrm{par}}} = 0.5$ m s$^{-1}$ and $\sigma_{w'} = 0.17$ m s$^{-1}$ would translate to a corresponding error in the amplitude of the $u'_{\mathrm{par}}$ and $w'$ oscillations. This in turn would lead to $\mathrm{MF_x}$ errors of $0.5/5 = 10\%$ and $0.17/1 = 17\%$, respectively. Assuming that these errors are random and uncorrelated, the $\mathrm{MF_x}$ error reduces in proportion to the number of samples through the wave. Considering waves with a length of $8$ km and larger, each wave is sampled with at least $10$ points, reducing the relative error by a factor of $F = 10^{-1/2} \approx 0.32$. Hence, the relative errors in $\mathrm{MF_x}$ are about $3.2\%$ and $5.5\%$, where the true error probably lies in between. For the following discussion, a relative $\mathrm{MF_x}$ error of $4\%$ is considered.

Moreover, as already demonstrated by Brown (1983), a further uncertainty of the leg-averaged $\mathrm{MF_x}$ is induced by the leg-length itself and a potentially related unequal sampling of updrafts and downdrafts. To cope with that issue, $\mathrm{MF_x}$ is calculated for several sub-legs with a fixed start point at the westernmost point of the track and a varying leg length. The minimum leg length is $3/4L$ km, and is stepwise increased by $1$ km intervals until the maximum leg length of $L$ is reached. The same procedure is additionally applied starting from the easternmost point of the flight track. Afterwards, the mean and standard deviation for all sub-legs is calculated, whereas the determined standard deviation represents the sensitivity of the leg-averaged $\mathrm{MF_x}$ to the start/end points and the length of the flight leg.

The derived horizontal wind along flight direction $u_{\mathrm{par}}$, the vertical wind speed as well as $u'_{\mathrm{par}}\,w'$ for the short waves ($\lambda < 30$ km) and the long waves ($\lambda > 30$ km) measured on FL3 are shown in Fig. 8. For $u'_{\mathrm{par}}\,w'$ shown in panels (c) and (d), only the altitude range between $7.5$ km and $10$ km with almost full data coverage is depicted for a better visibility of particular features.

The horizontal wind speed along the flight track $u_{\mathrm{par}}$ (Fig. 8, a) was measured to vary between about $30$ m s$^{-1}$ in the eastern part of the flight leg ($10°$ E to $15°$ E) and about $8$ m s$^{-1}$ in the western part ($7°$ E to $10°$ E). The obvious large-scale wave structures have a wavelength of about $400$ km with upstream tilted phase lines in the troposphere. The Falcon in-situ measurements depicted by the thin line at flight level in the tropopause region indicate similar wave structures but with higher wind speeds ($\approx 5$ m s$^{-1}$) compared to the tropospheric values. In addition to that, the large-scale wave structures are superimposed by small-scale waves with vertical phase lines, which are more clearly visible in the vertical wind measurements as they are shown in panel (b) of Fig. 8.

The vertical wind speed data looks comparable to the one measured on FL2 (see also Fig. 5), which confirms the stable atmospheric conditions during the flight period. The vertical wind speed varies between $-3$ m s$^{-1}$ and $3$ m s$^{-1}$ and shows horizontal scales in the range between $5$ and $30$ km. Thus, the spectral properties of the observed horizontal and vertical wind speeds are in-line with what was seen from GW-LCYCLE I data (Witschas et al., 2017), which revealed that the vertical wind speed spectrum mainly represents the short-wave spectrum, whereas the spectrum of the horizontal wind speed perturbations is dominated by the long-wave part but additionally shows an influence on the shorter wavelengths. Similar observations have been made by Smith and Kruse (2017) based on airborne in-situ data. Furthermore, the horizontal scales of the waves in the tropopause region, for instance in the upper range gates of the 2-$\mu$m DWL data ($9.0$ to $9.2$ km), as well as in the in-situ data at flight level ($9.8$ km), are significantly smaller ($\approx 5$ km) compared to the one in the troposphere. A similar behavior was already observed during FL2 (Fig. 5) and points to trapped and downstream propagating waves in the TIL region.

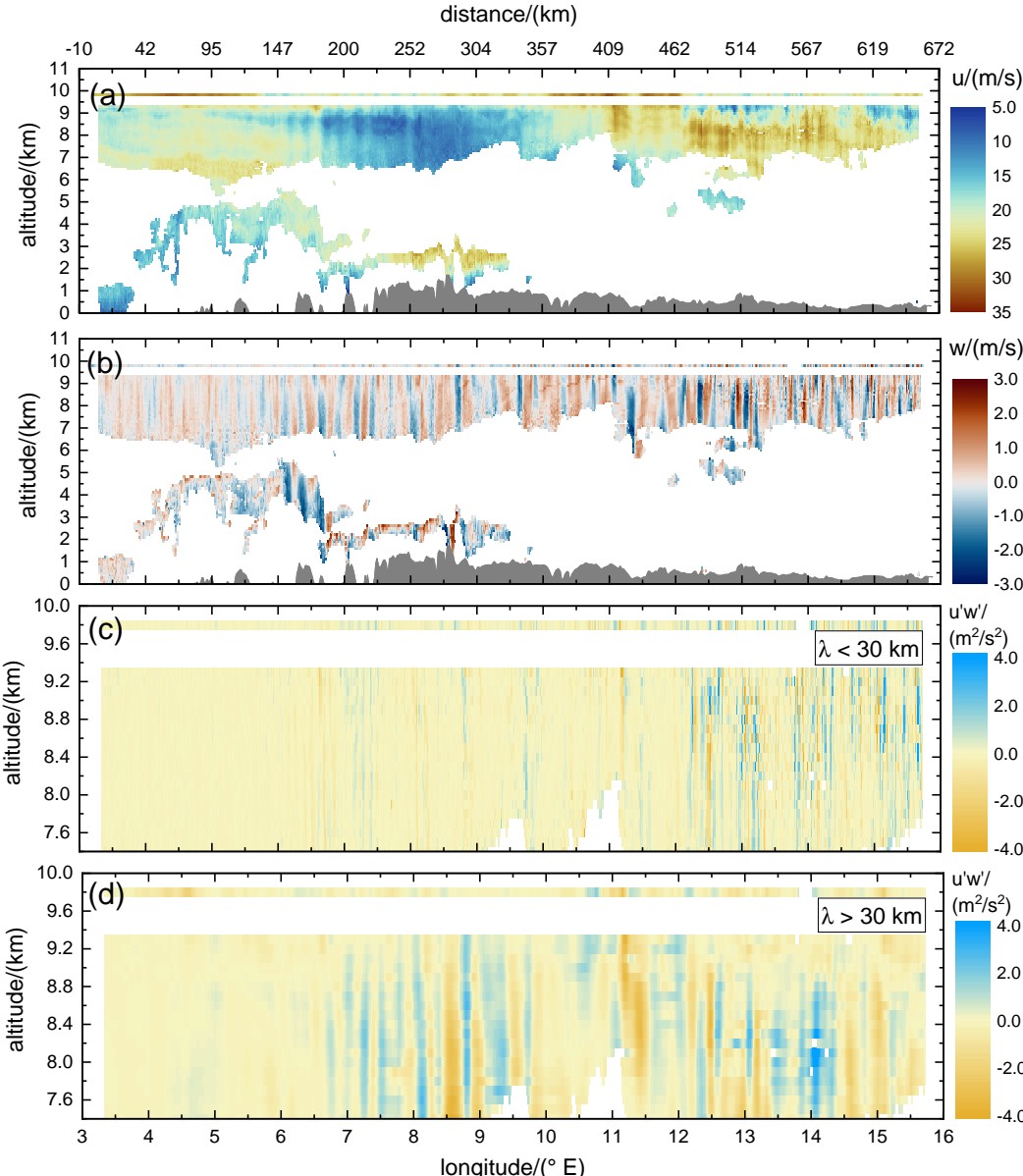

**Figure 8.** Horizontal wind speed along flight direction $u_{\mathrm{par}}$ (a), vertical wind speed (b), as well as $u'_{\mathrm{par}}w'$ for the short waves with a wavelength smaller than 30 km (c) and longer than 30 km (d) retrieved from the 2-$\mu$m DWL while operating in MF-mode on FL3 performed on 28 January 2016 (see also table 1, and Fig. 1). The thin line at 9.8 km altitude indicates the corresponding wind speed measured in-situ by the nose-boom of the aircraft. The orography is denoted by the gray area. For reasons of clarity, only the altitude range between 7.5 km and 10 km with almost full data coverage is depicted in panel (c) and (d).

From panel (c), it is obvious that the short-wave $u'_{\mathrm{par}} w'$ yields amplitudes of $\approx \pm 4 \, \mathrm{m}^2 \, \mathrm{s}^{-2}$, and that they are mainly visible in the downstream region ($12°$ E to $15.5°$ E). It can also be recognized that the amplitudes are larger for altitudes above the tropopause ($\approx 8.8 \, \mathrm{km}$), with alternating positive and negative signals of similar amplitude. Thus, the leg-averaged MF of these short waves is expected to be close to zero as shown later in Fig. 9.

For the long-wave $u'_{\mathrm{par}} w'$ (d), similar amplitudes of $\approx \pm 4 \, \mathrm{m}^2 \, \mathrm{s}^{-2}$ are reached, however, starting further upstream at around $7°$ E. Additionally, it can be realized that, as opposed to the short-wave class, the amplitudes get significantly smaller at the tropopause ($\approx 8.8 \, \mathrm{km}$) and above, except for the feature appearing at $11.4°$ E. Furthermore, in the tropopause region between $8.0$ and $8.8 \, \mathrm{km}$, the positive values of $u'_{\mathrm{par}} w'$ predominate, which gives reason to expect positive momentum fluxes, pointing to wave reflection. That these features are restricted to the tropopause region is for instance nicely visible at $12.5°$ E and $13.0°$ E, where two distinct positive $u'_{\mathrm{par}} w'$-features extend from $8.0$ and $8.8 \, \mathrm{km}$ whereas they are positive or rather zero below and above.

From the range-resolved measurements of $u'_{\mathrm{par}} w'$ (Fig. 8, c and d), the profile of the leg-averaged vertical flux of horizontal momentum $\mathrm{MF_x}$ is derived according to Eq. (3) and shown in Fig. 9 for the two different wave classes with $\lambda < 30 \, \mathrm{km}$ (orange) and $\lambda > 30 \, \mathrm{km}$ (blue). The mean MF retrieved from the 2-$\mu$m DWL data is denoted by the solid lines, the error bars denote the corresponding uncertainty which is approximated to be $4\%$ of the mean value, and the shaded area indicates the uncertainty induced by the varying leg length. The in-situ measurements are indicated by the dots in the respective altitude (see also figure label).

It is worth mentioning, that the presented method to derive the zonal MF only works reliably in the case of having unidirectional flow conditions along the entire flight leg, as only the wind component along the flight track is measured. For the shown example case, it was verified using ECMWF-IFS data, that $u_{\mathrm{par}}$ deviated from the horizontal wind speed by a maximum of $\pm 2 \, \mathrm{m} \, \mathrm{s}^{-1}$ in the analyzed altitude range ($7.8$ to $9.8 \, \mathrm{km}$), and thus, by less than 10%. Furthermore, the uncertainty induced by these slight changes, caused by a small wind direction change along the flight leg in different altitudes, is partly considered by the sequential leg integration as discussed before. To derive a reliable profile of the total MF, a modified scan pattern with fore/aft and additionally left/right steering laser beams is foreseen for future applications, to measure both, the horizontal wind component parallel to the flight direction $u_{\mathrm{par}}$ as well as perpendicular to the flight direction $u_{\mathrm{per}}$.

The flux profile of the short waves (orange) is undulating around zero due to the phase relationship of their horizontal and vertical velocity perturbations. Thus, GWs with a wavelength of smaller than $30 \, \mathrm{km}$ can be concluded to not transport momentum vertically. On the other hand, the flux profile of the long waves exhibits prominent features. Below $8 \, \mathrm{km}$, $\mathrm{MF_x}$ is slightly negative, indicating an upward propagation, whereas $\mathrm{MF_x}$ is positive between $8.0 \, \mathrm{km}$ and $8.8 \, \mathrm{km}$ in the tropopause region, suggesting downward propagation. This points to a reflection or rather trapping of the MWs in the vicinity of the TIL region. The largest $\mathrm{MF_x}$ values of $\approx 0.05 \, \mathrm{Pa}$ are found at an altitude of $8.2 \, \mathrm{km}$.

Furthermore, for the long wave analysis it can be seen that the $\mathrm{MF_x} \approx -0.1 \, \mathrm{Pa}$ derived from HALO in-situ measurements at $7.8 \, \mathrm{km}$ altitude (magenta) differs significantly from the $\mathrm{MF_x} \approx -0.02 \, \mathrm{Pa}$ derived from the 2-$\mu$m DWL (blue), which is surprising, as the respective horizontal and vertical wind speed measurements are in great accordance (see also Fig. 6). To

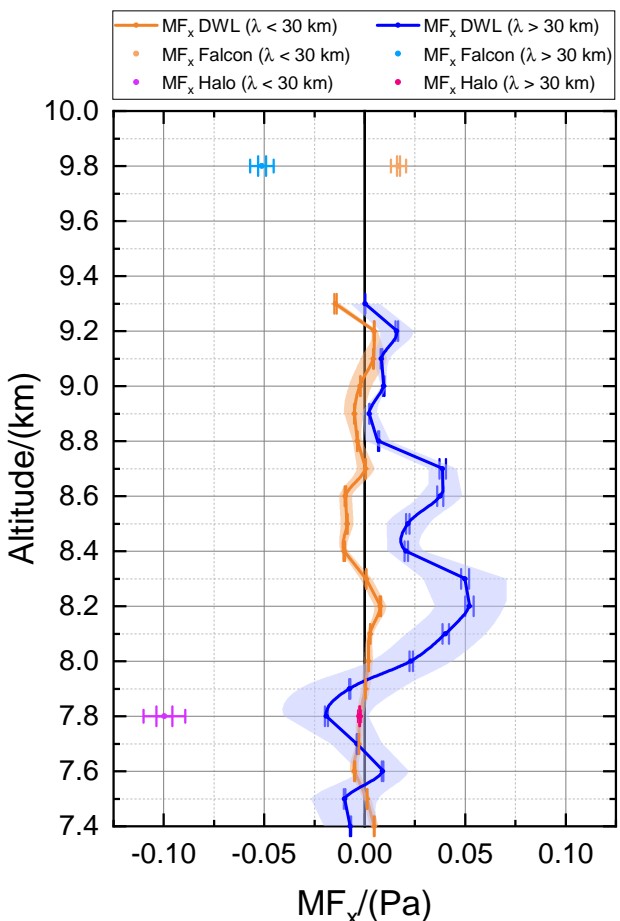

**Figure 9.** Leg-averaged MF profile calculated according to Eq. (3) for two different wave classes with $\lambda < 30$ km in orange colors and $\lambda > 30$ km in blue colors. The mean MF retrieved from the 2-$\mu$m DWL data is denoted by the solid lines, the error bars denote the corresponding uncertainty which is approximated to be $4\%$ of the mean value, and the shaded area indicates the uncertainty induced by the varying leg length. The in-situ measurements are indicated by the dots in the respective altitude.

investigate the root-cause of this discrepancy, the $u'_{par} w'$ data for the 2-$\mu$m DWL (black) and HALO (red) are compared for the short wave case and the long wave case as shown in Fig. 10 a, and b, respectively. For the short wave case, both time series are undulating around zero and with similar amplitudes, thus, providing comparable leg-averaged momentum fluxes. For the long-wave case, however, the $u'_{par} w'$ patterns look comparable but show distinct deviations between $8.5°$ E and $10°$ E, and 390 even more pronounced between $13°$ E and $15.5°$ E, where the 2-$\mu$m DWL data tendentially provides more positive values compared to the HALO in-situ data. Though the root-cause of these deviations is not unequivocal proven, it is likely that they are caused by the discrepancy between both data sets that appear between $14°$ E and $15.5°$ E, where the HALO ground speed

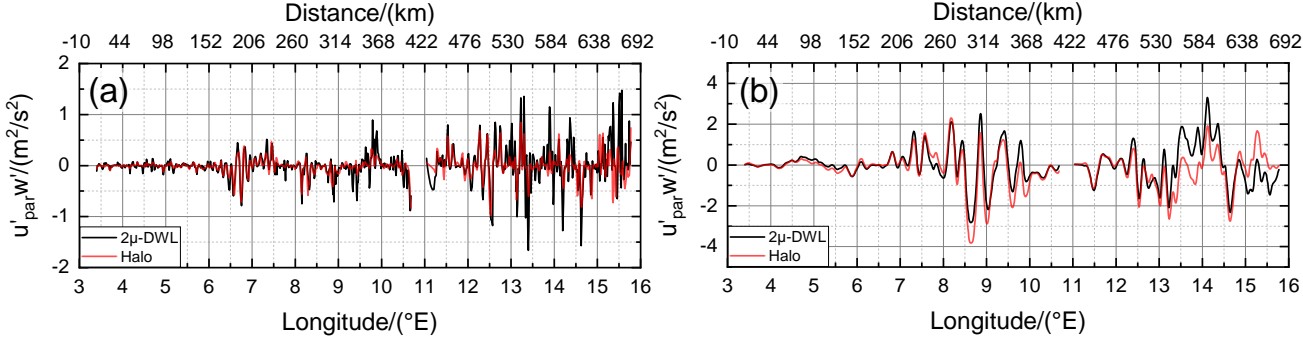

**Figure 10.** $u'_{\mathrm{par}}\, w'$ measured by the 2-$\mu$m DWL (black) and HALO (red) at 7.8 km for the short-wave class (a) and the long-wave class (b), respectively.

and attitude is shown to change (see also Fig. 6). As the 2-$\mu$m DWL ground returns in this region do not show an enhanced systematic error, the 2-$\mu$m DWL results are considered to be reliable here (see also Fig. 7).

## 5 Summary

In the framework of the GW-LCYCLE II campaign, which took place from 12 January 2016 to 3 February 2016 in northern Scandinavia, the DLR 2-$\mu$m DWL was deployed on the DLR Falcon research aircraft and was, together with other airborne and ground-based instruments, used to investigate the properties of orographically induced GWs over the Scandinavian mountain ridge. To extend the capabilities of the 2-$\mu$m DWL to measure the vertical wind and the horizontal wind along flight direction simultaneously with an enhanced horizontal resolution of $\approx 800$ m, a novel MF scanning mode (MF-mode), with alternately fore and aft propagating laser beams was applied on a cross-mountain flight leg performed on the second research flight on 28 January 2016. As this flight leg was coordinated with the HALO aircraft flying at lower altitudes of 7.8 km, HALO in-situ measurements could be used to validate the functionality of MF-mode measurements and the corresponding retrieval algorithms. It is shown that the derived vertical wind speeds have a mean systematic error of only $0.01$ m s$^{-1}$ and a corresponding random error of $0.17$ m s$^{-1}$. For the horizontal wind speed measured along flight direction, the systematic error is $0.76$ m s$^{-1}$ and the random error is $0.50$ m s$^{-1}$, whereas the systematic error is composed of the 2-$\mu$m DWL contribution and the one of the HALO measurements which is specified to be $\approx 0.6$ m s$^{-1}$.

Horizontal wind speed measurements performed by the 2-$\mu$m DWL along the Scandinavian coast are shown to be useful to characterize the overall inflow conditions. They revealed that the wind was blowing with up to $30$ m s$^{-1}$ in the region where the Falcon cross-mountain leg was performed ($\approx 61°$ N) and that wind direction was more or less perpendicular to the orientation of the mountain ridge in all altitudes (from the ground up to $9.8$ km).

Vertical wind speed measurements with a higher horizontal resolution of about $200$ m clearly show the excitation of GWs. The amplitudes of the vertical wind speed reach values of up to $\pm 3$ m s$^{-1}$ and the phase lines are shown to be vertically

orientated. Furthermore, a change in the horizontal wavelength is observed in the downwind region ($5°$ N to $8°$ N) and altitudes of about $8$ km. As shown by Gisinger et al. (2020), this behavior can be explained by the occurrence of interfacial waves with wavelengths smaller than $10$ km, that are induced by the partial reflection of longer waves at the TIL region. Currently, further numerical simulations are ongoing to investigate if the generation and downward propagation of secondary waves in the stratosphere also contribute to the observed phenomenon.

Based on MF-mode data, the leg-averaged horizontal momentum profile was derived for two different wave classes with wavelengths smaller than $30$ km and for wavelengths larger than $30$ km. The shorter waves are shown to not transport any momentum. On the other hand, for the long waves, a negative flux of $\approx -0.02 \, \mathrm{m}^2 \, \mathrm{s}^{-2}$ was determined at an altitude of $7.8$ km, indicating an upward GW propagation. On contrary, $\mathrm{MF_x}$ was positive above $8$ km up to $8.8$ km, suggesting a downward propagation. A fact which points to a reflection or trapping of MWs in the TIL region.

Hence, this analysis demonstrates that $2$-$\mu$m DWL measurements, in general, are beneficial compared to in-situ measurements as they provide similar accuracy and a sufficient horizontal resolution, but wind information in several altitudes enables to study the GW propagation processes in more detail. Changes of GW properties in dynamically interesting regions such as the tropopause are vertically well resolved which cannot be captured by in-situ observations. Even if multiple stacked legs with $100$ m vertical spacing would be possible, they would suffer from the time shift among them. The MF-mode measurements are particularly useful, as they further allow to characterize the MF profile and with that, to gain knowledge about the propagation direction of the excited GWs. An adaption of the presented MF-mode measurements by the observation of the horizontal wind component perpendicular to the flight direction ($u_{\mathrm{per}}$) is foreseen for future applications to be able to derive the total MF profile.

*Author contributions.* Benjamin Witschas prepared the main part of the paper manuscript, developed the retrieval for the MF from $2$-$\mu$m DWL data, and performed the corresponding analysis. Stephan Rahm performed the $2$-$\mu$m DWL data processing and provided the line-of-sight wind speeds and lidar housekeeping data enabling to perform of the MF retrieval. Sonja Gisinger provided an overview of the meteorological conditions during the entire campaign period and contributed to discussions about the physical interpretation of the retrieved MF profiles. Markus Rapp was the principal investigator of the GW-LCYCLE II campaign and contributed to the preparation of the manuscript. Andreas Dörnbrack and Dave Fritts supported the development of the novel GW-scan pattern and contributed to the preparation of the paper manuscript.

*Competing interests.* The authors declare that they have no conflict of interest.

*Acknowledgements.* The technical assistance by Engelbert Nagel as well as the support of the DLR flight facility for the realization of the performed validation campaigns is highly acknowledged.

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
