# Peer review of "Airborne coherent wind lidar measurements of the momentum flux profile from orographically induced gravity waves"

_Atmospheric Measurement Techniques, 2022_

## Author Comment (AC1)

**(Author response)**

**Reviewer statement:**

The article reports on mountain-wave momentum-flux (MF) measurements performed during the GW-LCYCLE II airborne campaign above Scandinavia. A novel scanning pattern technique, which has been specifically designed to retrieve gravity-wave momentum fluxes, was used with the 2µm lidar flown onboard the Falcon aircraft during one leg flight across the Scandinavian mountains on January 28, 2016. This technique is based on classical radar MF measurements, where the beam is pointed obliquely in two opposite directions, which allows to measure both the wind vertical component and its horizontal component along the line of sight. During that leg, the HALO aircraft performed a coordinated flight, flying 2 km below the Falcon aircraft, therefore allowing a direct comparison first between lidar and in-situ winds, and then between gravity-wave momentum fluxes. While the former comparison is excellent, the later is somewhat disappointing.

The article is well written and presents a very promising technique for measuring gravity wave MF from an airborne platform. The detailed 2D perspective that it provides on MF is notably really impressive. I therefore support the publication of the article in AMT. Yet, I have several concerns on the paper, notably on how the gravity-wave observations are (or are not) interpreted, and ultimately on the significance of the momentum flux comparison. I think that these concerns need to be addressed before publication.

**Major concerns**

My main major concern is associated with Figure 8a, where gravity waves MF derived from the 2µm wind lidar are shown for different altitudes. One striking feature of these fluxes is that they present oscillations around 0. This feature is simply ignored in the paragraph devoted to the figure (lines 269-287), while it has profound implications on the type of waves that are observed. Indeed, one can imagine two situations associated with MFs that oscillate around 0: it may either be associated with freely-propagating gravity-wave packets with systematical horizontal direction of propagation almost perpendicular to the aircraft leg (which seems rather unlikely in this mountain-wave case), or it may be trapped waves for which u' measured along the wave propagation direction and w' disturbances are in phase quadrature. This latter situation seems to me very plausible in the considered case: it is for instance consistent with the (almost) vertical structure of the short-scale wave disturbances displayed on Figure 7c) and d). I also note that the authors very briefly proposed the same interpretation on lines 312-313 (while commenting Fig. 9) and in the conclusions (lines 342-343). Now, if most of the waves observed during the aircraft leg were trapped waves, one would expect that the associated leg-averaged MF should nearly vanish. I therefore wonder whether leg-averaged momentum fluxes shown in Figure 9 are not simply residuals from the almost zero-mean timeseries displayed in Figure 8a), which might explain the observed discrepancy between the lidar and in-situ MF estimates. In other words, despite the excellent agreement between both wind measurements, this leg might not be the best case to compare gravity-wave momentum fluxes.

Thanks a lot for this comment. As a few of the shown results were already presented in Gisinger et al. 2021, we originally aimed at discussing the novel measurement technique and the corresponding retrieval algorithm, instead of elaborating on the actual atmospheric mechanism that triggers the observed phenomenon. However, we fully agree that the scientific discussion of the prevailing gravity wave situation was not detailed enough in the previous version of the manuscript. Thus, we carefully revised our analysis and the corresponding discussion presented in Section 4. In particular, we performed the following "major" modifications:

-   As suggested by Ehard et al. 2015, we applied a Butterwort filter to retrieve the respective perturbations in u' and w'. In particular, this was done for two different wave classes with horizontal wavelengths of smaller than 30 km, and for horizontal wavelengths of larger than 30 km. With that, it is possible to distinguish between the contribution of the longer wavelengths that are able to propagate through the troposphere and can transport momentum and the one of the shorter wavelengths that do not transport momentum vertically at all. With that, we address both valuable comments of the referee about the actual gravity wave situation in the TIL region and the propagation direction, as well as the mentioned discrepancy with the applied filter method used to retrieve u' and w'.
-   With the two defined wave classes, we adapted the previous Fig. 7 according to:

[Figure]

**Figure 8.** Horizontal wind speed along flight direction $u_{par}$ (a), vertical wind speed (b), as well as $u'_{par} w'$ for the short waves with a wavelength smaller than 30 km (c) and longer than 30 km (d) retrieved from the 2-$\mu$m DWL while operating in MF-mode on FL3 performed on 28 January 2016 (see also table 1, and Fig. 1). The thin line at 9.8 km altitude indicates the corresponding wind speed measured in-situ by the nose-boom of the aircraft. The orography is denoted by the gray area. For reasons of clarity, only the altitude range between 7.5 km and 10 km with almost full data coverage is depicted in panel (c) and (d).

Based on the shown observations, we further characterize the potential gravity wave situation that can explain our observations and we looked into more detail into the difference that was found in MF between HALO and the wind lidar by means of u'w':

- Based on the vertical wind speed measurement shown in Fig. 5:

   Analyzing the spectral structure of the excited GWs, an interesting behavior can be observed between 5° E to 8° E (see also the light-gray line in Fig. 5). Below altitudes of ≈ 8.5 km, and thus below the thermal tropopause, which is determined using European Centre for Medium-Range Weather Forecasts (ECMWF) model data, the waves have a horizontal wavelength of about 20 km, whereas the ones in or rather above the tropopause have a horizontal wavelength of only 10 km and smaller. As shown by Gisinger et al. (2020), this behavior can be explained by the occurrence of interfacial waves with wavelengths smaller than 10 km, that are induced by the partial reflection of longer waves at the TIL region. Currently, further numerical simulations are ongoing to investigate if the generation and downward propagation of secondary waves in the stratosphere also contribute to the observed phenomenon.

- Based on the u, w and u'w' 2D plots:

   The horizontal wind speed along the flight track upar (Fig. 8, a) … The obvious large-scale wave structures have a wavelength of about 400 km with upstream tilted phase lines in the troposphere. … … . In addition to that, the large-scale wave structures are superimposed by small-scale waves with vertical phase lines, which are more clearly visible in the vertical wind measurements as they are shown in panel (b) of Fig. 8.

   The vertical wind speed data looks comparable to the one measured on FL2 (see also Fig. 5), which confirms the stable atmospheric conditions during the flight period. The vertical wind speed varies between −3 m/s and 3 m/s and shows horizontal scales in the range between 5 and 30 km. … Furthermore, the horizontal scales of the waves in the tropopause region, for instance in the upper range gates of the 2-µm DWL data (9.0 to 9.2 km), as well as in the in-situ data at flight level (9.8 km), are significantly smaller (≈ 5 km) compared to the one in the troposphere. A similar behavior was already observed during FL2 (Fig. 5) and points to trapped and downstream propagation waves in the TIL region.

   From panel (c), it is obvious that the short-wave u' par w' yields amplitudes of ≈ ±4 $m^2/s^2$, and that they are mainly visible in the downstream region (12° E to 15.5° E). It can also be recognized that the amplitudes are larger for altitudes above the tropopause (≈ 8.8 km), with alternating positive and negative signals of similar amplitude. Thus, the leg-averaged MF of these short waves is expected to be close to zero as shown later in Fig. 9.

   For the long-wave u' par w' (d), similar amplitudes of ≈ ±4 $m^2/s^2$ are reached, however, starting further upstream at around 7◦ E. Additionally, it can be realized that, as opposed to the short-wave case, the amplitudes get smaller at the tropopause (≈ 8.8 km) and above, except for the feature appearing at 11.4◦ E. Furthermore, in the tropopause region between 8.0 and 8.8 km, the positive values of u' par w' predominate, which gives reason to expect positive momentum fluxes, pointing to the reflection of waves. That these features are restricted to the tropopause region is for instance nicely visible at 12.5◦ E and 13.0◦ E, where two distinct positive u' par w'-features extend from 8.0 and 8.8 km whereas they are positive or rather zero below and above.

- Based on the MF profile shown in Fig. 9:

   The flux profile of the short waves (orange) is undulating around zero due to the phase relationship of their horizontal and vertical velocity perturbations. Thus, GWs with a wavelength of smaller than 30 km can be concluded to not transport momentum vertically. On the other hand, the flux profile of the long waves exhibits prominent features. Below 8 km, MFx is slightly negative, indicating an upward propagation, whereas MFx is positive between 8.0 km and 8.8 km in the tropopause region, suggesting downward propagation. This points to a reflection or rather

trapping of the MWs in the vicinity of the TIL region. The largest MFx values of ≈ 0.05 Pa are found at an altitude of 8.2 km.
- new Fig. 10 shows u'w' of HALO in-situ and wind lidar along the flight leg for small (<30km) and large (>30km scales) to shed light on the differences found in the MF:

[Figure]

**Figure 10.** $u'_{par}\,w'$ measured by the 2-$\mu$m DWL (black) and Halo (red) at 7.8 km for the short-wave case (a) and the long-wave case (b), respectively.

For the short-wave case, both time series are undulating around zero and with similar amplitudes, thus, providing comparable leg-averaged momentum fluxes. For the longwave case, however, the $u'_{par}w'$ patterns look comparable but show distinct deviations between 8.5° E and 10° E, and between 13° E and 15.5° E, where the 2-$\mu$m DWL data tendentially provides more positive values compared to the Halo in-situ data. Though the root-cause of these deviations is not unequivocal proven, it is likely that they are caused by the discrepancy between both data sets that appear between 14° E and 15.5° E, where the HALO ground velocity and attitude is shown to change (see also Fig. 6). As the 2-$\mu$m DWL ground returns in this region do not show an enhanced systematic error, the 2-$\mu$m DWL results are considered to be reliable here (see also Fig. 7).

Somehow related to the previous comment, I have another concern associated with the filtering that is chosen to extract the wave disturbances from the raw observations. On line 252, it is stated that a "5th order polynomial" is used to determine the background wind. On a 700-km long leg, this will typically filter out wavelengths longer than 150 km. On the other hand, the authors note on line 257, while commenting Figure 7a), that the u_par wind varies with wavelength of about 400 km, as is also obvious in Figure 6 (upper right panel). What was the reason to filter out this wavelength? It might actually be that this wavelength is not associated with a trapped wave and might therefore be a better testbed for lidar and in-situ MF comparisons. This can be achieved by choosing a different filter to extract the fluctuations from the background, e.g. a simple straight line between end points of the raw observations.

On the other hand, I do not agree that this 400-km wavelength is also appearing in Figure 7b), as stated in l 258. It should have been filtered out from the disturbances!

Thanks a lot for this comment. Indeed, not only a 5th order polynomial was used for background subtraction, but an additional low pass filter. Realizing that, we carefully re-processed the data and used only a Butterworth filter for two defined wave classes for short waves (< 30 km) and long waves (> 30 km). Based on that, we revised the u'w' figures, the corresponding wavelet analysis, and the discussion about the retrieved results as described above.

**Other concerns**

▪ The article discusses the same measurements than those studied in Gisinger et al. (2020, ref. cited), and shares a number of very similar figures (e.g. Fig 5, 7 and 9 in this paper, compared to Fig. 8, 9 and 10 in Gisinger et al.(2020)). This may be acknowledged since the Introduction, where the focus of this paper with respect to Gisinger et al. might be stressed.

 - The present paper intended to concentrate on the novel measurement technique procedure of the momentum flux profiles (AMT) whereas the paper by Gisinger et al., used a much broader data set and also simulations to perform a more science-related study. For completeness and also as a measurement example of the usefulness of the presented method, we decided to partly keep this content also in this publication but additionally acknowledged this both in the abstract as well as in the introduction. In particular, after defining the two different wave classes, we base all our discussion on the 2D-plot of u, w, and u'w', but exclude the additional wavelet analysis as it does not provide further conclusions. Furthermore, we keep the flux profile but calculate it for the two different wave classes. The publication of Gisinger et al is additionally acknowledged:

 • Abstract: …, which are induced by interfacial waves as recently presented by Gisinger et al. 2020.
 • Introduction: Whereas this paper concentrates on the description of the novel measurement technique and the careful characterization of related uncertainties based on in-situ measurements, the scientific results based on the retrieved leg-averaged momentum flux profile have partly been published by Gisinger et al. 2020 but are kept in this paper for completeness.

▪ l. 37: This sentence is slightly confusing: only the projection of gravity-wave MFs *on the flight direction* can be estimated. In other words, the "par" direction is that of the flight, not that of the different wave packets. This probably needs to be reminded to the reader more explicitly. Of course, in the mountain-wave case presented here, the leg direction has been chosen to be along the expected direction of propagation of mountain wave packets (which might also be more explicitly highlighted). Thanks a lot for the hint. For further clarification, we added the following sentence:

 …as this would allow one to additionally quantify the propagation direction of the GWs and the corresponding momentum transport, especially when the flight leg direction is chosen such that it matches the expected propagation direction of excited GWs.

▪ I had difficulties in understanding the reasoning behind the "wind mode" and the "vertical wind speed" modes of the lidar, since 3D winds are already measured with the first mode (as far as I have understood). My understanding is that the horizontal resolution and the accuracy/precision of the retrieved vertical wind speed is different in both modes, but this is not explicated at first place in the paper. I would therefore recommend to provide further details on the two modes as soon as line 95-97, when lidar modes are first mentioned: e.g. wind mode means 3D wind vector with lower horizontal resolution than in the vertical mode. Writing this, I am yet not fully sure if the "Wind vector" data product reported in Table 1 is a 3D or 2D (horizontal) vector.

 - Though the different 2-µm DWL scanning modes are explained in detail in the respective section, we fully agree that further explanation is needed earlier in the manuscript, to provide a better understandability. Thus, we added the following sentences to the manuscript:

 • Sec.2.2: …with the 2-µm DWL operating in wind-mode measuring both wind speed and wind direction with a horizontal resolution of ≈ 8.4 km and a vertical resolution of 100 m, hence, giving the possibility to investigate the inflow conditions based on the measured wind speed and wind direction.

 • On this leg, the 2µ DWL performed in vertical-wind-mode where the laser beam is pointed to nadir-direction to measure the vertical wind speed with a horizontal resolution of 200 m and a vertical resolution of 100 m (see also Sect. 3.2), which enables to resolve the small-scale structure of the excited MWs.

- Commonly, the 2-$\mu m$ DWL is used to either measure the three-dimensional wind vector (wind-mode) or rather to measure the vertical wind speed (vertical-wind-mode). When operating in wind-mode, the velocity-azimuth display (VAD) scan technique is applied (Browning and Wexler, 1968) by performing a conical scan around the vertical axis with an off-nadir angle of 20°. Typically, one scanner revolution with 21 line-of-sight (LOS) measurements separated by 18° in the azimuth direction takes 175 about 42 s. By further considering the aircraft speed of about 200 m s-1, the horizontal resolution of wind-mode observations is about 8.4 km, depending on the actual ground speed of the aircraft. Hence, the vertical velocity wind field, which is shaped by scales of tens of kilometers and smaller in the presence of MWs (Smith and Kruse, 2017), is not well resolved in wind mode data.

- l. 154: Related to the previous comment, this sentence is also confusing. "Simultaneous measurements of the horizontal and vertical wind speed" are already achieved in the "wind mode" if a 3D wind vector is retrieved. The advantage of the new scan pattern seems to me more associated with the horizontal resolution and the precision of the measurements rather than in their simultaneity.
  - We fully agree that it is not really the simultaneity but the higher horizontal resolution and the reduced representativeness error of the MF-mode measurements which make them beneficial. We clarified that by the previous and following adaptions:
  - However, it was also discussed that simultaneous measurements of the horizontal and the vertical wind speed with a horizontal resolution of a few hundred meters and a low representativeness error would be even more beneficial as such measurements would allow retrieving the vertical flux of horizontal momentum induced by GWs (Smith et al., 2008, 2016).

- l. 160-161: "flying along wind direction": I guess you had in mind the special case of mountain wave (in a homogeneous wind field). Either extend why it is important to fly along the wind direction here, or simply remove this since the MF scan technique does not request to fly along the *wind* direction. (see also my comment for l 37.)
  - We agree that the MF-mode measurements do not require a flight path along the in-flow direction. Thus, the sentence "and flying along wind direction," was removed. The particular explanation why the flight leg direction was planned along the wind direction for the MW event is given later in chapter 4.

- Equations 2 just look wrong to me. Starting from Eq. (1), I obtain different formulas:
  u_par = csc(theta_f - theta_b) * ( v_f cos(theta_b) - v_b cos(theta_f))
  w = csc(theta_f - theta_b) * (-v_f sin(theta_b) + v_b sin(theta_f))
  - Thanks a lot for reviewing Equations (1) and (2) so carefully as indeed two typos were present in Equation (2). Hence, Equation (2) was corrected according to:
  $$u_{\mathrm{par}}(x, R) = \csc(\theta_{\mathrm{f_i}} - \theta_{\mathrm{b_i}})(v_{\mathrm{f_i}} \cos \theta_{\mathrm{b_i}} - v_{\mathrm{b_i}} \cos \theta_{\mathrm{f_i}})$$

  $$w(x, R) = \csc(\theta_{\mathrm{f_i}} - \theta_{\mathrm{b_i}})(-v_{\mathrm{f_i}} \sin \theta_{\mathrm{b_i}} + v_{\mathrm{b_i}} \sin \theta_{\mathrm{f_i}})$$
  Further, it was verified that all the data processing was performed with the correct equations.

- Figure 5: Why are you using a different x-axis in this figure (4 to 17°E) and in the following ones (3 to 16°E)?
  - Fig. 5 shows the vertical wind measurement from FL2, whereas Fig. 6 and Fig. 7 show the result from the MF-mode measurements acquired on FL3. As both flight legs do not have an identical length (see also Fig. 1) it is decided to change the x-axes scale to keep the maximum resolution. Hence, we would suggest keeping the x-axis scales as they are.

- ▪ Figure 8: I would recommend to reverse the rows in the figure in order to ease comparisons with Figure 7 for instance: i.e., put the top/bottom altitudes in the top/bottom panel.
  - As explained in the beginning, we decided to remove the figure about the wavelet analysis as it does not provide additional information for the presented discussion and as it is mostly presented already in Gisinger et al., 2020.

- ▪ Figure 8b: Since MF is a quadratic quantity, it varies horizontally with a wavelength that is half that of the wave-packet u' and w' disturbances. The interpretation of Figure 8b) in terms of "wavelengths" of the wave packet is therefore a bit confusing.
  - This figure and corresponding discussion was removed.

**Typos and minor concerns**
- ▪ l 28: Did GW-LCYCLE II campaign occurred in 2014? or in 2016 (see e.g. caption of Figure 1)??
  - It was in 2016, and the typo in the text was adapted accordingly.
- ▪ l. 178: the data *are* linear*ly* interpolated.
  - Adopted.
- ▪ l 226-228: Please refer to Figure 6 here.
  - Done.

---

## Author Comment (AC2)

**Reviewer statement:**

The paper by B. Witschas and co-authors presents a method to measure gravity wave momentum fluxes from line-of-sight wind observations using an airborne 2-micron Doppler wind Lidar (DWL). The technique is applied to measurements gathered over a flight leg of the Falcon aircraft during the GW-LCYCLE II campaign in Scandinavia (2016). Lidar wind and momentum flux retrievals are compared with colocated in situ measurements by the HALO aircraft available for this case study. While the manuscript is well-written and the topic of interest to AMT readership, a major point of criticism I have is that a significant part of the material presented was already published by two of the authors of the present manuscript in Gisinger et al. (2020, article cited). Some figures are very similar to that previous study. In this context, I would expect the present manuscript to provide a thorough description of the instrument and its performance. On the contrary, although the method is sound and the results very impressive, the technical discussion remains superficial, in particular regarding the advantages of this new measurement mode with respect to other scanning modes and the uncertainties of the technique. I appreciate that earlier papers by the authors may already provide some of this information, but it would be necessary to repeat some of the details here. For those reasons, the paper should be reconsidered after major revisions.

Thanks a lot for this comment. Indeed, the primary goal of this paper is to address the measurement technique, corresponding algorithms, and reached accuracies, rather than discussing the scientific results as was done by Gisinger et al., 2020. Still, we decided to keep at least the MF-flux profile plot as an application example and differences between HALO in-situ and lidar are investigated in more detail (new Fig. 10). Furthermore, we added a whole paragraph about the LOS wind retrieval and added the analysis of the lidar ground return signals which indirectly demonstrated how accurate the measurements (and the flight attitude control loop for the scanner) work. Furthermore, to reduce repetitions with the work presented by Gisinger et al., 2022, we removed the figure with the wavelet analysis, as it does not provide additional information for the discussion. A detailed listing of the performed changes in the manuscript is given below.

**Major comments :**

1) I find that the paper lacks an a priori estimate of wind error and resulting momentum flux noise level. Granted, the agreement with in situ measurements is very good (for wind), but there may be sources of difference other than instrumental errors (e.g., slightly different timing inducing a phase shift). Would it not be possible to estimate the expected error, even roughly, and compare with the empirical estimate? This point should at least be discussed.

Thanks a lot for raising that point. As shown by the direct comparison of the Lidar and the in-situ data for u and w, the slight temporal and spatial occurrence of the respective measurements do not induce a

significant error. However, small deviations are obvious, especially in the range between 14°E and 15.8°E. We realized that, in this region, the HALO ground speed and with that, flight attitude changed significantly by a pitch angle change from 1.4° to 2.0°. Also, for the Falcon, the pitch angle changed in that region. To verify if this has affected our lidar measurement accuracy, we additionally analyzed the ground returns which should result in a LOS wind speed of 0 m/s. And indeed, no significant discrepancies from 0 m/s could be observed. On the other hand, the differences between Lidar and Halo in-situ data could be correlated with the observed changes in the HALO pitch angle. Thus, it is likely that the in-situ measurements are slightly affected by the changing flight attitude.

Regarding the ground return analysis, we added the following figure and the corresponding paragraph:

Figure 7. Line-of-sight wind speed retrieved from 2-µm DWL ground returns (black) and moving average of 10 successive data points (orange).

In contrast to the in-situ measurements, airborne wind lidar measurements enable to estimate the accuracy of the performed wind observations by analyzing the returns from the non-moving ground which should yield values of 0 m/s by definition. For the discussed flight leg, ground returns are available from about  $9.5^{\circ}$  E to  $15.8^{\circ}$  E as shown in Fig. 7, black. A moving average of 10 successive data points is additionally indicated by the orange line. It can be seen that the LOS wind speeds are nicely varying around 0 m/s with peak-to-peak variations mostly below 0.1 m/s. The mean of this data set yields a value of 0.00 m/s with a standard deviation of 0.05 m/s, further demonstrating the high accuracy of the 2-µm DWL measurements during this flight. Furthermore, the flight attitude changes between 14° E to  $15.5^{\circ}$  E do not induce any remarkable systematic error, confirming that the discrepancy of 2-µm DWL and HALO in-situ data is unlikely to be caused by the lidar measurements.

In addition, we added an estimate of the MF uncertainty based on the random errors of u and w according to Smith et al., 2016. Here, it turns out that the error induced by the not defined flight leg length is larger than the one originating from the random error of the lidar measurements. In order to clarify this situation, we added the following paragraph to the manuscript:

The uncertainty of the derived momentum fluxes can be estimated as suggested by Smith et al. (2016). As the mean values are removed before calculating MFx, the corresponding uncertainty of MFx only arises from the random errors of  $u_{par}$  and w. To propagate these errors for the momentum flux, a transect with anti-correlated sinusoidal u' par and w' oscillations is considered with common amplitudes of 5.0 m s-1 and 1.0 m s-1, respectively. Furthermore, a mean air density of  $\rho = 0.5$  kg m-3 for the altitude range between 7.0 km and 10.0 km is assumed. This enables to define a reference value of the leg-averaged momentum flux MFx =  $\rho < u'_{par}w' > = (0.5)(0.5)(5.0)(1.0) = 1.25$  Pa. In a worst-case situation, the random errors  $\sigma u'_{par} = 0.5$  m s-1 and  $\sigma w' = 0.17$  m s-1 would translate to a corresponding error in the amplitude of the  $u'_{par}$  and w' oscillation. This would lead to MFx errors of 0.5/5 = 10% and 0.17/1 = 17%, respectively. Assuming that these errors are random and uncorrelated, the MFx error reduces in proportion to the number of samples through the wave. Considering waves with a length of 8 km and larger, each wave is sampled with at least 10 points, reducing the relative error by a factor of F =  $10-1/2 \approx 0.32$ . Hence, the relative errors in MFx are about 3.2% and 5.5%, where the true error probably lies in between these two values.

Furthermore, the authors do not explain the preliminary steps involved in LOS wind retrievals (e.g., estimating Doppler shift, subtracting the aircraft ground-relative speed). The potential impact of aircraft motions and associated uncertainty could also be discussed in more detail.

Thanks a lot for this comment. Originally, it was planned to just concentrate on the MF-mode retrieval, as the other processing steps were explained in detail in previous publications for instance in Chouza et al. 2016 and Witschas et al., 2017. But we agree that further information is helpful for the reader, especially in a measurement-technique-related paper such as this one. For this reason, we extended the instrument description with a condensed explanation of the LOS processing steps. The impact of the aircraft motions and other noise sources is furthermore characterized by means of the ground return analysis that was also added to the manuscript (Fig. 7 of the new manuscript and corresponding explanation). Section 3.1 was extended by the following paragraph:

.... harsh environment of an aircraft. To do so, the beat signal of the emitted laser pulse and the local oscillator (seed laser) is analyzed. When the beat frequency or the laser pulse built-up time exceeds a certain difference from their nominal values, the signal from the respective laser pulse is not considered for accumulation. Moreover, before accumulating the respective reference pulse spectra, they are frequency-shifted to a defined reference value to correct for pulse-to-pulse frequency variations and thus avoid spectral broadening in the accumulation process. A similar frequency shift is also applied to the atmospheric signal power spectra. Afterwards, the signals from the remaining laser pulses in a one-second time interval are spectrally averaged. The part of the detector raw signal containing the atmospheric return is divided into segments that lead to 100-m range gates in the vertical by considering the actual laser beam pointing angle, the aircraft altitude and attitude, and the reference pulse timing. After that, the power spectrum is calculated for each range gate and laser pulse, is frequency shifted according to the reference pulse frequency shift and, subsequently, accumulated. The detector signal at the end of the record (below the ground return) is used to analyze and correct the detector noise characteristics. The power spectrum of the noise signal is calculated for each single laser pulse and is additionally averaged for a one-second time interval. Consequently, the power spectrum for each single range gate is divided by the respective noise spectrum to correct for the system noise and the receiver frequency response (noise whitening). In the next step, the power spectra are corrected for the aircraft speed projected onto the LOS direction, which is derived from the ground speed measured by the GPS module and the actual laser beam pointing direction. The latter is calculated from the actual scanner position, the aircraft attitude (inertial reference system) as well as the lidar installation position with respect to the aircraft reference frame (Chouza et al., 2016). Finally, the LOS wind speed  $v_{LOS}$  is calculated according to  $v_{LOS} = (\Delta f c)/(2 f_0)$ , where  $\Delta f$  is the frequency shift between the reference pulse and the atmospheric signal,  $f_0$  is the laser frequency, c is the velocity of light, and  $\lambda_0 = c/f_0 = 2022.54$  nm is the laser wavelength. To retrieve vertical or horizontal wind information from respective LOS wind measurements, further processing steps are needed, as discussed in Sect. 3.2.

2) The comparison between Lidar and in situ wind observations could be more thorough. For instance, what is the power spectral density of the differences in Fig. 6? Are there specific artifacts at given frequencies?

The analysis of the respective u' and w' power spectra did not reveal any significant differences between in-situ and lidar measurements. As mentioned above, we determined small deviations in the range between 14°E and 15.8°E and realized that the HALO aircraft pitch angle changed from 1.4° to 2.0° in this region. Thus, we could not determine given frequencies that may explain the differences between lidar and in-situ measurements but could allocate the region where the differences occur. Furthermore, the analysis of the 2- $\mu$ m DWL ground returns gives further confidence that the lidar measurements are accurate, as the shown deviations in the range of only 0.05 m/s. Moreover, we have split our analysis in two different wave classes of short waves (smaller than 30 km) and long waves (larger than 30 km). By doing so, it can be shown that the discrepancy of the retrieved MF values is only true for the long wave

class. To further address that issue and to visualize where the actual deviations between both data sets appear, we added a u'w' plot for both data sets and both wave classes as shown below:

Figure 10.  $u'_{\text{par}} w'$  measured by the 2- $\mu$ m DWL (black) and Halo (red) at 7.8 km for the short-wave case (a) and the long-wave case (b), respectively.

**The following paragraph was also added to the manuscript:**

Furthermore, for the long wave analysis it can be seen that the MFx  $\approx -0.1$  Pa derived from HALO in-situ measurements at 7.8 km altitude (magenta) differs significantly from the MFx  $\approx -0.02$  Pa derived from the 2-µm DWL (blue), which is surprising, as the respective horizontal and vertical wind speed measurements are in great accordance (see also Fig. 6). To investigate the root cause of this discrepancy, the u' par w' data for the 2-µm DWL (black) and Halo (red) are compared for the short-wave class and the long wave class as shown in Fig. 10 a, and b, respectively. For the short-wave case, both time series are undulating around zero and with similar amplitudes, thus, providing comparable leg-averaged momentum fluxes. For the long-wave case, however, the u'par w' patterns look comparable but show distinct deviations between 8.5° E and 10° E, and between 13° E and 15.5° E, where the 2-µm DWL data tendentially provide more positive values compared to the Halo in-situ data. Though the root cause of these deviations is not unequivocal proven, it is likely that they are caused by the discrepancy between both data sets that appear between 14° E and 15.5° E, where the Halo ground speed and attitude are shown to change (see also Fig. 6). As the 2-µm DWL ground returns in this region do not show an enhanced systematic error, the 2-µm DWL results are considered to be reliable here (see also Fig. 7).

**Other comments :**

Please double check Eq. 2. I obtain the same result as Referee 1, different from yours. Thanks a lot for reviewing Equations (1) and (2) so carefully as indeed two typos were present in Equation (2). Hence, Equation (2) was corrected according to:  $(a, B) = \cos(\theta_{1}, \theta_{2})(a, \cos\theta_{2}, w) = \cos(\theta_{1})$

 $u_{\text{par}}(x,R) = \csc(\theta_{f_{i}} - \theta_{b_{i}})(v_{f_{i}}\cos\theta_{b_{i}} - v_{b_{i}}\cos\theta_{f_{i}})$

 $w(x,R) = \csc(\theta_{f_{i}} - \theta_{b_{i}})(-v_{f_{i}}\sin\theta_{b_{i}} + v_{b_{i}}\sin\theta_{f_{i}})$

Furthermore, it was verified that all the data processing was performed with the correct equations.

Line 148-149 : Could you elaborate a bit on the 'scanner control loop on a 1-second basis'? What is the uncertainty in attitude and how does it translate in LOS wind uncertainty ? Are pilot oscillations of attitude present? If yes, at which frequency? Are they sufficiently resolved at 1 s?

Thanks a lot for the question. The flight attitude in general changes only slightly in a one-second time frame and it depends on the actual pointing direction in the attitude changes and how this will translate into a LOS error. The best indication of the accuracy of retrieved LOS winds for airborne wind lidars is the analysis of ground returns (over land) if available. During the discussed research flight we were lucky and had a certain number of usable ground returns (see figure below). These measurements demonstrate a

mean LOS wind speed for the ground returns of 0 m/s and a random error of 0.05 m/s. Furthermore, the peak-to-peak fluctuation is more or less always less than 0.1 m/s, which means that potential attitude oscillations within the one-second time interval lead to an error of less than 0.1 m/s. The flight legs are flown with auto-pilot.

---

## Author Response (AR2)

**(Author response)**

**Reviewer statement:**
I was acting as Reviewer #1 on the previous version of the paper. I would like to thank the authors for taking into account all the remarks of the reviewers. The improvement of the paper has been, in my opinion, substantial. I, therefore, support its publication, with only a few technical details that need to be corrected:

- l 111: you probably rather want to provide the longitude of the Falcon turn, since FL3 took place at 61°N altogether.
  Indeed, it makes much more sense to mention the longitude position here.
  61° N was replaced by 3° E

- Eq 1: there should not be a negative sign in the vbi equation. Indeed, if one assumes that theta_bi = -theta_fi, as suggested on the following line, equations for vfi and vbi would be similar! Eq. (2), which are used for the wind retrieval, are correct as long as there is no negative sign in the vbi equation too.
  Thanks again for another detailed verification of this equation. Of course, this is correct and was changed accordingly in the manuscript.

- Fig 6d): the insert indicates a negative bias, whereas the mean value looks positive on the figure. I suspect a sign error. If so, please also correct l. 281.
  This is absolutely correct. The bias calculated as the difference between the Halo in-situ winds and the DWL measurements is positive. This was corrected accordingly in the figure insert as well as in the text (line 281). The absolute value was proven to be correct.